# Targeted two-photon chemical apoptotic ablation of defined cell types *in vivo*

Robert A. Hill[1,2], Eyiyemisi C. Damisah[1,3], Fuyi Chen[1,2], Alex C. Kwan[2,4] & Jaime Grutzendler[1,2]

A major bottleneck limiting understanding of mechanisms and consequences of cell death in complex organisms is the inability to induce and visualize this process with spatial and temporal precision in living animals. Here we report a technique termed two-photon chemical apoptotic targeted ablation (2Phatal) that uses focal illumination with a femtosecond-pulsed laser to bleach a nucleic acid-binding dye causing dose-dependent apoptosis of individual cells without collateral damage. Using 2Phatal, we achieve precise ablation of distinct populations of neurons, glia and pericytes in the mouse brain and in zebrafish. When combined with organelle-targeted fluorescent proteins and biosensors, we uncover previously unrecognized cell-type differences in patterns of apoptosis and associated dynamics of ribosomal disassembly, calcium overload and mitochondrial fission. 2Phatal provides a powerful and rapidly adoptable platform to investigate *in vivo* functional consequences and neural plasticity following cell death as well as apoptosis, cell clearance and tissue remodelling in diverse organs and species.

[1] Department of Neurology, Yale School of Medicine, New Haven, Connecticut 06511, USA. [2] Department of Neuroscience, Yale School of Medicine, New Haven, Connecticut 06510, USA. [3] Department of Neurosurgery, Yale School of Medicine, New Haven, Connecticut 06511, USA. [4] Department of Psychiatry, Yale School of Medicine, New Haven, Connecticut 06511, USA. Correspondence and requests for materials should be addressed to J.G. (email: jaime.grutzendler@yale.edu).

Experimental approaches for cell ablation have been important tools for investigating a variety of biological questions. However, applications of cell ablation in living organisms, especially in complex mammalian systems, have been limited due to a lack of methods able to precisely induce and image the death process of individual cells *in vivo*. Ideally, these methods would have precise temporal and spatial specificity, and hijack intrinsic apoptotic cellular mechanisms to mimic the *in vivo* situation. Numerous pharmacological agents lacking spatiotemporal precision are available that can induce widespread apoptotic cell death in culture and *in vivo*[1–3]. On the other hand, methods for more targeted cell death induction have been developed but they have practical and technical limitations that restrict their utility for studies in living animals. These methods fall into three general categories: chromophore assisted light inactivation (CALI)[4–10], two-photon thermal ablation[11–14] and genetically encoded death receptors[15–17]. Some of their fundamental limitations include a need to efficiently and accurately deliver the required dyes or genetic materials for targeted cell killing, prohibitively long illumination requirements, non-specific tissue damage and necrotic death with spilling of cellular contents, and induction of rapid macrophage/microglial activation. These limitations have precluded precise real-time *in vivo* molecular and cellular studies of single-cell apoptosis in complex mammalian organisms. As a result, there remain significant gaps in the understanding of the physiological consequences, multicellular reactions and tissue plasticity that occur after cell death in various organs.

To overcome these issues, we have developed a powerful and rapidly adoptable method for induction of apoptosis in single cells of interest in living organisms. This method, which we termed 2Phatal (two-photon chemical apoptotic targeted ablation), uses a femtosecond-pulsed laser to induce highly focal photo-bleaching of a nuclear-binding dye. This leads to dose-dependent single-cell apoptosis, likely to be due to DNA damage caused by bleaching-induced reactive oxygen species (ROS) production. Combined with high-resolution time-lapse imaging, 2Phatal constitutes, to our knowledge, the first targeted *in vivo* single-cell apoptosis platform that is robust, reproducible and amenable to precise cell biological analysis and quantification. Using this method, we demonstrate in the live mouse brain, induction of apoptosis in neurons, astrocytes, NG2 glia and vascular pericytes, and in zebrafish neuromast lateral line hair cells. In combination with genetically encoded subcellular organelle labelling and calcium biosensors, we identify unique cell-type-dependent differences in the temporal profile of cell death and a novel sequence of ribosomal disassembly, calcium overload and mitochondrial fission never before visualized *in vivo*. Finally, we also provide multiple proof-of-principle studies, which open up the possibility of further dissection of mechanisms and functional consequences of cell ablation in the intact *in vivo* system by testing the consequences of ablating a small group of fast spiking interneurons on the excitability of a local cortical circuit. Thus, 2Phatal opens a range of capabilities for the comprehensive interrogation in living organisms of apoptotic death pathways, multicellular glial reactions associated with cell death and circuit-based consequences of targeted cell removal.

## Results

**Targeted photochemical induction of cell death *in vivo*.** Methods based on photo-bleaching of fluorescent molecules to induce ROS production have been extensively used to inactivate organelle function and kill cells *in vitro*[7,10,18,19]. We thus reasoned that to induce targeted single-cell killing *in vivo*, we could cause bleaching of fluorophores within individual cells of interest using the focal illumination properties of a conventional two-photon femtosecond pulsed laser. However, when we used this approach in the live mouse brain, we found that focal bleaching of fluorescent proteins or the fluorophore sulforhodamine 101 (SR101) in neurons or astrocytes was ineffective at causing their death even at higher laser intensities (Supplementary Fig. 1). Even when fluorescent proteins were targeted to the nucleus to potentially cause more bleaching-induced DNA damage, we did not observe cell death (Supplementary Fig. 1). To overcome this limitation, we reasoned that photobleaching of a dye that is in closer proximity and tightly interacting with nuclear DNA could be an effective way to induce ROS-mediated DNA damage that would trigger the apoptosis cascade leading to death of single cells. We thus administered to live mice the cell permeant nuclear-binding dye Hoechst 33342 (H33342) by topical application to the cortex through a cranial window or by intravenous systemic injection (Supplementary Fig. 2). Within 3 h, both administration methods extensively labelled nuclei, which were visible at least 300 µm deep into the cortex with excellent signal to noise ratio (Supplementary Movie 1). Application of H33342 in mice expressing yellow fluorescent protein in neuronal subsets (*Thy1-YFP*) (Fig. 1b) or concurrently with the dye SR101, which can label astrocytes[20] (Fig. 1b and Supplementary Movie 2), showed that nuclei of both cell types were robustly labelled (astrocytes were 2.1-fold brighter than neurons but there was minimal variability in labeling within cell types. $P = 0.015$, 60 astrocytes and 60 neurons from $n = 3$ mice, Student's *t*-test) (Fig. 1c). Dye labelling caused no detectable cellular toxicity under normal imaging conditions and was cleared from the tissue within days (Supplementary Fig. 2). Thus, H33342 can be used for both acute and chronic experimental preparations and is compatible with repeated *in vivo* imaging.

To test whether we could induce targeted cell killing *in vivo*, regions of interest (ROIs) ($8 \times 8$ µm) were centred on single cells and briefly scanned using the two-photon laser (tuned to 775 nm) to cause dye bleaching (see Methods, Fig. 1d). In contrast to bleaching of non-nuclear dyes, time-lapse imaging revealed that H33342 caused a stereotyped progression of nuclear pyknosis, fragmentation and cell disappearance starting $\sim 2$ h after photobleaching (Fig. 1d). Importantly, unlike previous methods of two-photon-mediated laser ablation[11,13,14,21–24], this mild, brief photobleaching approach did not induce cell rupture or tissue burning (Fig. 1). To develop a dose-dependent, easily adoptable and highly reproducible method for single-cell killing, we systematically tested the effects of laser scan time and laser intensity on H33342 photobleaching and cell death progression. As predicted, a linear correlation was evident between laser scan time and H33342 photobleaching ($R^2 = 0.999$, $n = 30$ cells per condition from three mice) (Fig. 1e,f). Repeated imaging of the targeted cells revealed that the rates of apoptotic cell death (as quantified by nuclear pyknosis and subsequent cell disappearance) were proportional to laser illumination ($n = 30$ cells per scan time from three mice) (Fig. 1g,h), as evidenced by a linear positive correlation between scan time ($R^2 = 0.9601$), total bleached units per cell ($R^2 = 0.9604$) and cell death induction (Fig. 1h and Supplementary Fig. 3). In addition to scan time, modulation of laser intensity also resulted in dose-dependent induction of cell death. Scan times were fixed at 10 s per cell and laser intensity was varied from 21.3 to 45.4 mW (measured at the sample plane). As predicted, increased laser intensities correlated with the extent of ROI photobleaching (Fig. 1i). More importantly, there was a laser intensity-dependent induction of cell death with 21.3 mW resulting in 50% and 40.6 mW resulting in 100% death ($n = 30$ cells per intensity from three mice) (Fig. 1j,k), positive correlations between laser

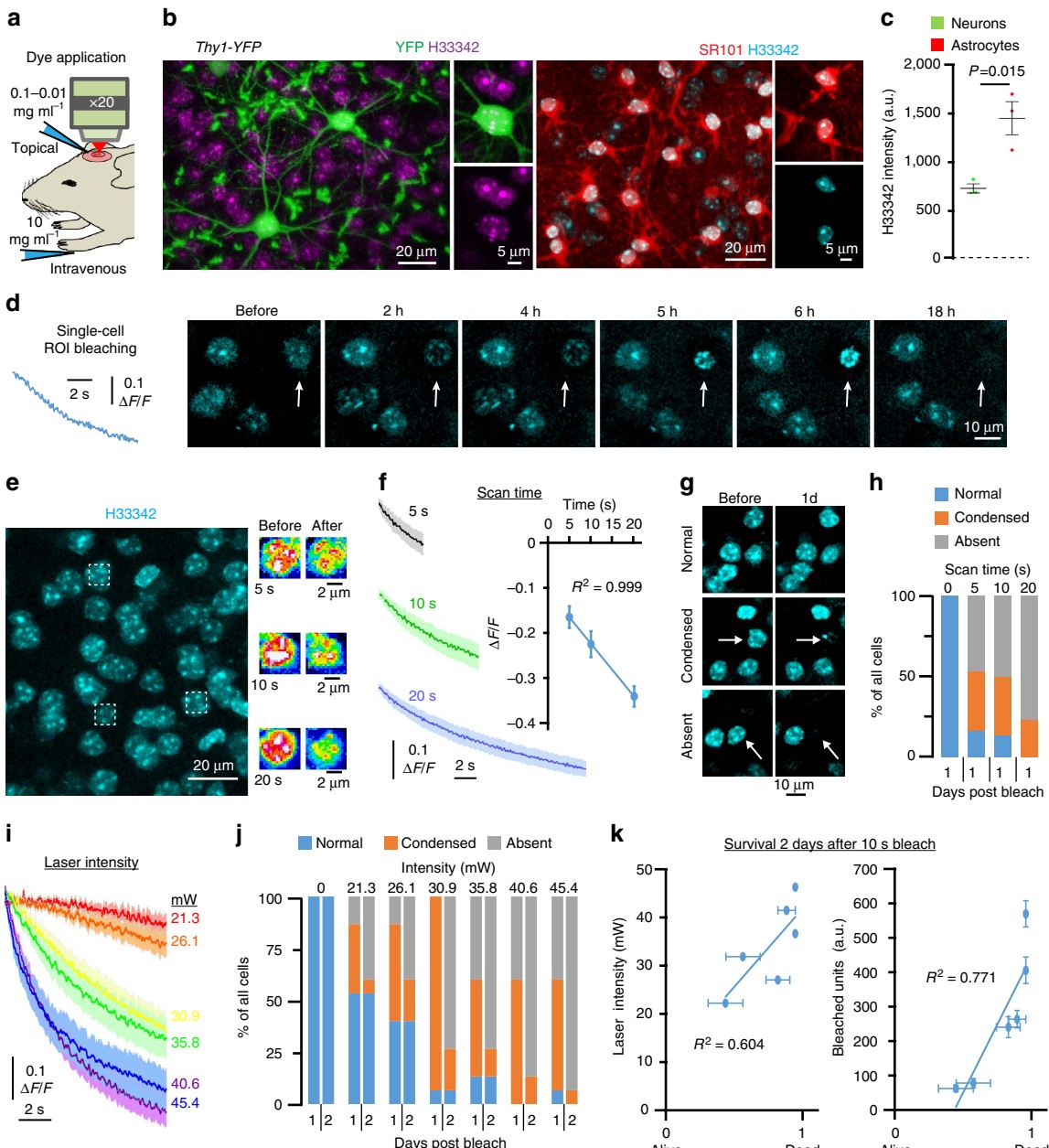

**Figure 1 | Two-photon photobleaching of nuclear-binding dye to ablate single cells *in vivo*.** (**a**) Schematic of *in vivo* imaging and labelling of the mouse cortex with Hoechst 33342 (H33342) via topical or intravenous dye application. (**b**) Images captured *in vivo* showing H33342 labelling in YFP-labelled neurons (green) and SR101-labelled astrocytes (red). (**c**) Box and whiskers plot showing H33342 fluorescence intensity is significantly higher in astrocytes compared to neurons. Sixty astrocytes and 60 neurons from $n = 3$ mice, t-test $P = 0.0151$, $t = 4.0783$, df = 4. (**d**) Fluorescence intensity trace showing photobleaching of H33342 within a single-cell ROI and subsequent nuclear condensation and disappearance of the targeted cell over 18 h (arrow). (**e**)Three cells (boxed regions) subjected to 5, 10 and 20 s of focal photobleaching. (**f**) Fluorescence intensity traces showing photobleaching of H33342 at the laser scan times indicated, traces indicate mean ± s.e.m. Linear correlation between laser scan time and single-cell bleaching. (**g**) Images showing the three cell death outcomes (arrows indicate targeted cells) quantified for each experimental condition. (**h**) Laser scan time-dependent cell death induction 1 day after photobleaching, $n = 30$ cells per intensity from 3 mice. (**i**) Fluorescence intensity traces showing photobleaching of H33342 at the laser intensities indicated, traces indicate mean ± s.e.m. (**j**) Laser intensity-dependent cell death induction 1 and 2 days after photobleaching, $n = 30$ cells per intensity from 3 mice. (**k**) Linear correlations between laser intensity and detection of cell death and units bleached per cell and cell death, traces indicate mean ± s.e.m.

intensity ($R^2 = 0.604$), total bleached units per cell ($R^2 = 0.771$) and cell death rates (Fig. 1k and Supplementary Fig. 3). Thus, fine modulation of laser scan time and intensity provides a robust approach for experimentally defined induction of cell death.

**2Phatal does not induce neural collateral damage.** A major obstacle associated with previous efforts to study the

consequences of cell removal stem from the often-overlooked tissue damage and responses by neighbouring glia that occur when using two-photon thermal laser ablation. For example, it is well documented that brain microglia respond to this kind of ablation via rapid process extension towards the injury site[25]. To test whether 2Phatal would induce a similar kind of acute response, microglia were visualized in *CX3CR1-GFP* reporter transgenic mice, whereas single neurons were

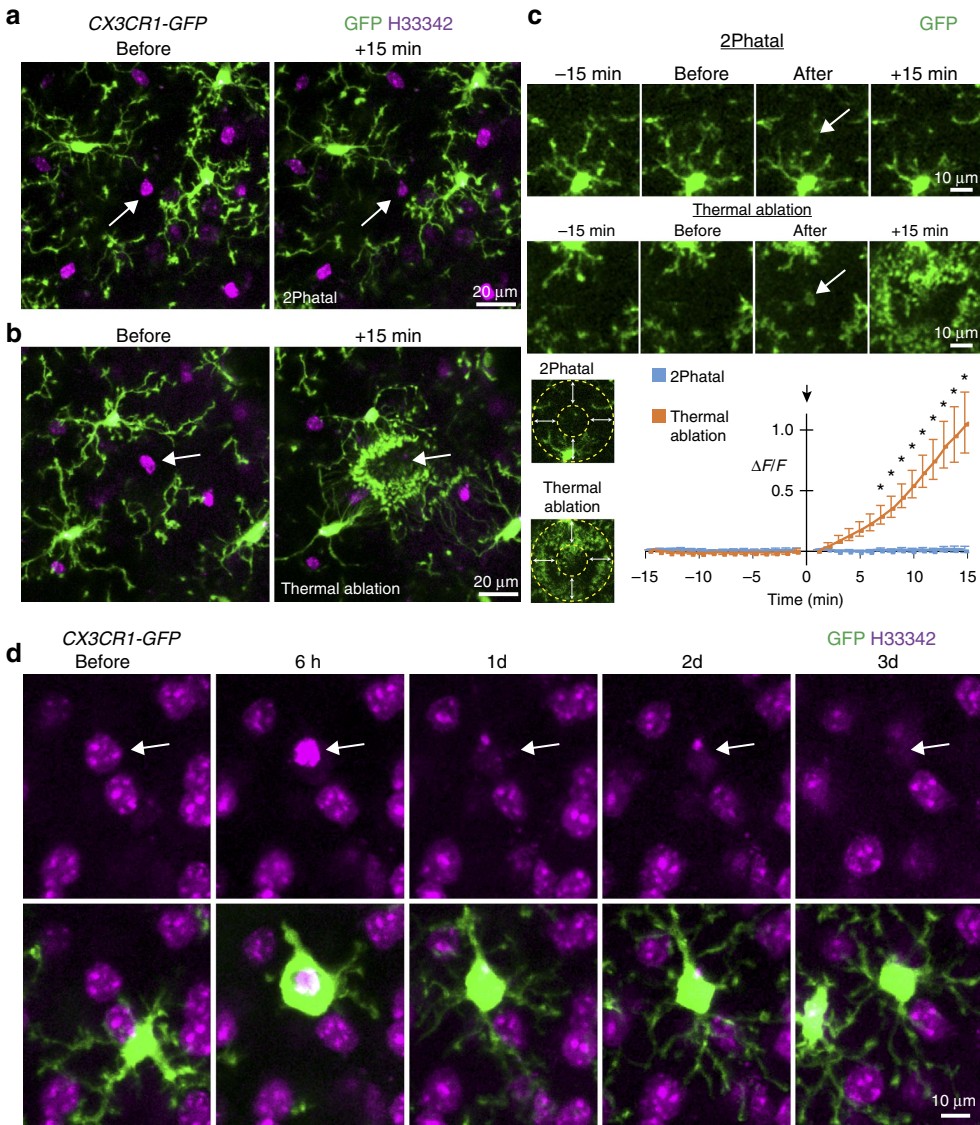

**Figure 2 | Microglial engulfment and phagocytosis of targeted cells.** (**a**) *In vivo* images showing *CX3CR1-GFP*-labelled microglia (green) adjacent to a cell before and after photobleaching (arrows), showing no change in microglia motility. (**b**) *In vivo* images showing microglia adjacent to a cell (arrows) before and after two-photon thermal ablation, showing rapid directed microglia process extension towards the mild thermal injury. (**c**) Time-lapse images and method for quantification of microglia response to 2Phatal and thermal ablation (arrows indicated targeted cell region). Unlike thermal ablation, there was no difference in microglia motility detected around cells subjected to 2Phatal; Thermal ablation $n = 7$ cells from 4 mice, 2Phatal $n = 9$ cells from 4 mice, traces indicate mean ± s.e.m. *Significance determined by a 99% confidence interval compared with baseline. (**d**) Extended time-lapse imaging revealed directed single-cell microglia phagocytosis of the apoptotic nuclei (arrow), providing the first system to study molecular and cellular mechanisms of single-cell apoptotic phagocytosis *in vivo* ($n = 12$ cells from 3 mice).

ablated with 2Phatal. *In vivo* time-lapse imaging revealed that in contrast to thermal ablation, 2Phatal did not induce any acute microglial process responses towards the targeted cell (Fig. 2a–c and Supplementary Movie 2) (thermal ablation $n = 7$ cells from 4 mice, 2Phatal $n = 9$ cells from 4 mice, significance* determined by a 99% confidence interval compared with baseline).

In contrast to the lack of an acute (within minutes) reaction by microglia to 2Phatal bleaching, subsequent imaging during the time points when nuclear pyknosis was occurring revealed a targeted process of single-cell phagocytosis by individual microglia ($n = 12$ cells from 3 mice) (Fig. 2d). Single microglia engulfed the dying cell and remained in the region for several days. This highly targeted, controlled response suggests precise mechanisms of microglia phagocytosis during single-cell

apoptosis and 2Phatal now provides a novel tool to study this process for the first time *in vivo*.

Next, to determine whether 2Phatal induced any collateral damage to nearby neuronal structures, single cells were ablated immediately adjacent to green fluorescent protein (GFP)-labelled axons and their boutons in *Thy1-GFP* transgenic mice (Fig. 3a). Time-lapse imaging over the following days revealed characteristic apoptotic nuclear condensation in the ablated cell but no significant effects on adjacent axonal boutons, which retained their normal plasticity rates compared to control regions away from ablation sites (42 cells for each condition from $n = 3$ mice, control vs 2Phatal, bouton formation: 1d $P = 0.981$, 2d $P = 0.960$; bouton elimination: 1d $P = 0.721$, 2d $P = 0.854$; stable 1d $P = 0.744$, 2d $P = 0.907$, multiple *t*-tests with Holm–Sidak correction) (Fig. 3a,b).

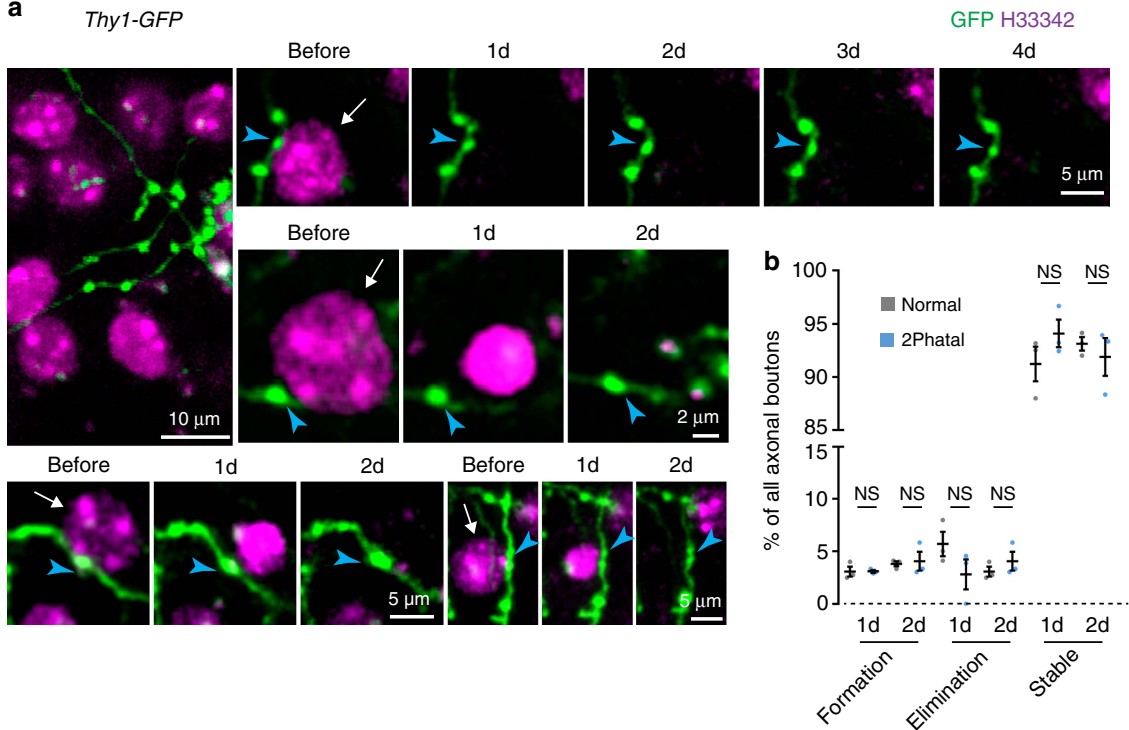

**Figure 3 | 2Phatal does not cause collateral damage to immediately adjacent neural structures.** (**a**) *In vivo* images showing GFP-labelled axonal boutons adjacent to H33342-labelled nuclei targeted for ablation with 2Phatal (arrows). During nuclear pyknosis, no significant changes in immediately adjacent boutons (blue arrowheads) was detected. (**b**) No differences in axonal bouton plasticity adjacent to normal or 2Phatal ablated cells over 2 days; 42 cells for each condition from $n = 3$ mice, $t$-test, control vs 2Phatal, formation: 1d $P = 0.981$, $t = 0.0249$, df = 4; 2d $P = 0.960$, $t = 0.2706$, df = 4; elimination: 1d $P = 0.721$, $t = 1.5700$, df = 4; 2d $P = 0.854$, $t = 0.9813$, df = 4; stable 1d $P = 0.744$, $t = 1.3843$, df = 4; 2d $P = 0.907$, $t = 0.6567$, df = 4 (df = degrees of freedom).

Thus, unlike previous ablation approaches, 2Phatal does not induce acute microglial process reactivity or disrupt immediately adjacent neuronal structures (Figs 2 and 3). Therefore, 2Phatal is the first available method that can be applied to reliably investigate at single-cell resolution how the surrounding neural microenvironment responds to a physiologically relevant mode of cell death, including phagocytosis of single apoptotic cells (Fig. 2d); all without the confounding factors of laser damage, spilling of cellular contents and rapid inflammatory glial responses.

**Cellular and molecular changes are consistent with apoptosis.**
Based on our finding of a stereotyped time course of nuclear condensation and cell death (Fig. 1), and the lack of acute microglia responses (Fig. 2), we hypothesized that 2Phatal induced a classical apoptotic process. To further verify this we characterized the morphological progression of cell death by visualizing cytoplasmic and membrane bound fluorescent reporters in both cortical pyramidal neurons (*Thy1-YFP*) and somatostatin-expressing interneurons (*SSTcre:mT/mG*). *In vivo* time-lapse imaging revealed that targeted cells proceeded through an apoptotic-like morphological cascade characterized by nuclear pyknosis and fragmentation coinciding with cell soma shrinkage, loss of dendrites and formation of apoptotic bodies, specifically at previous dendritic branch locations. (Fig. 4a–e, Supplementary Movie 3 and Supplementary Fig. 4). Neighbouring cells and processes showed no signs of collateral damage (Fig. 4d,e), similar to our quantifications of the effects of 2Phatal on axonal bouton plasticity (Fig. 3a,b).

The morphological characteristics of the death process suggested that 2Phatal was inducing classical cellular apoptosis.

However, to further validate this, we labelled dying cells *in vivo* with Annexin V, a protein which binds specifically to exposed phosphatidylserine in the cellular membrane of apoptotic cells. Indeed, fluorescently conjugated Annexin V was detected specifically on dying cell soma and processes as early as 2 h after nuclear photobleaching ($n = 15$ cells from 4 mice) (Fig. 4f), providing strong additional evidence for apoptosis induction. The ability to induce apoptosis in selected neuronal subtypes provides a powerful system to dissect the cell death process and to test the distinct contributions of targeted neuronal cell types to cortical network physiology *in vivo*.

**Calcium dynamics and mitochondrial fission during apoptosis.**
We next explored whether 2Phatal could be used to investigate subcellular, molecular and organelle dynamics during apoptosis. Calcium signalling is of fundamental importance during the apoptosis cascade[26,27]. We thus labelled neurons with the genetically encoded calcium indicator GCaMP6s via adeno-associated virus (AAV1) injected intrathecally (see Methods) or in *SSTcre:GCaMP6s* transgenic mice. Repeated *in vivo* time-lapse imaging in awake, head-fixed mice revealed that cells subjected to 2Phatal demonstrated a robust rise in intracellular calcium concentration characterized by sustained increases in GCaMP6 fluorescence in both cell bodies and dendrites (Fig. 5a,b and Supplementary Movie 4). The calcium overload occurred 2–3 h after H33342 photobleaching and coincided temporally with the initiation of nuclear morphological changes (Fig. 5c and Supplementary Fig. 4). Within minutes of the cytoplasmic calcium overload, GCaMP6s fluorescence could be observed transitioning from predominantly cytoplasmic labelling to both cytoplasmic and nuclear

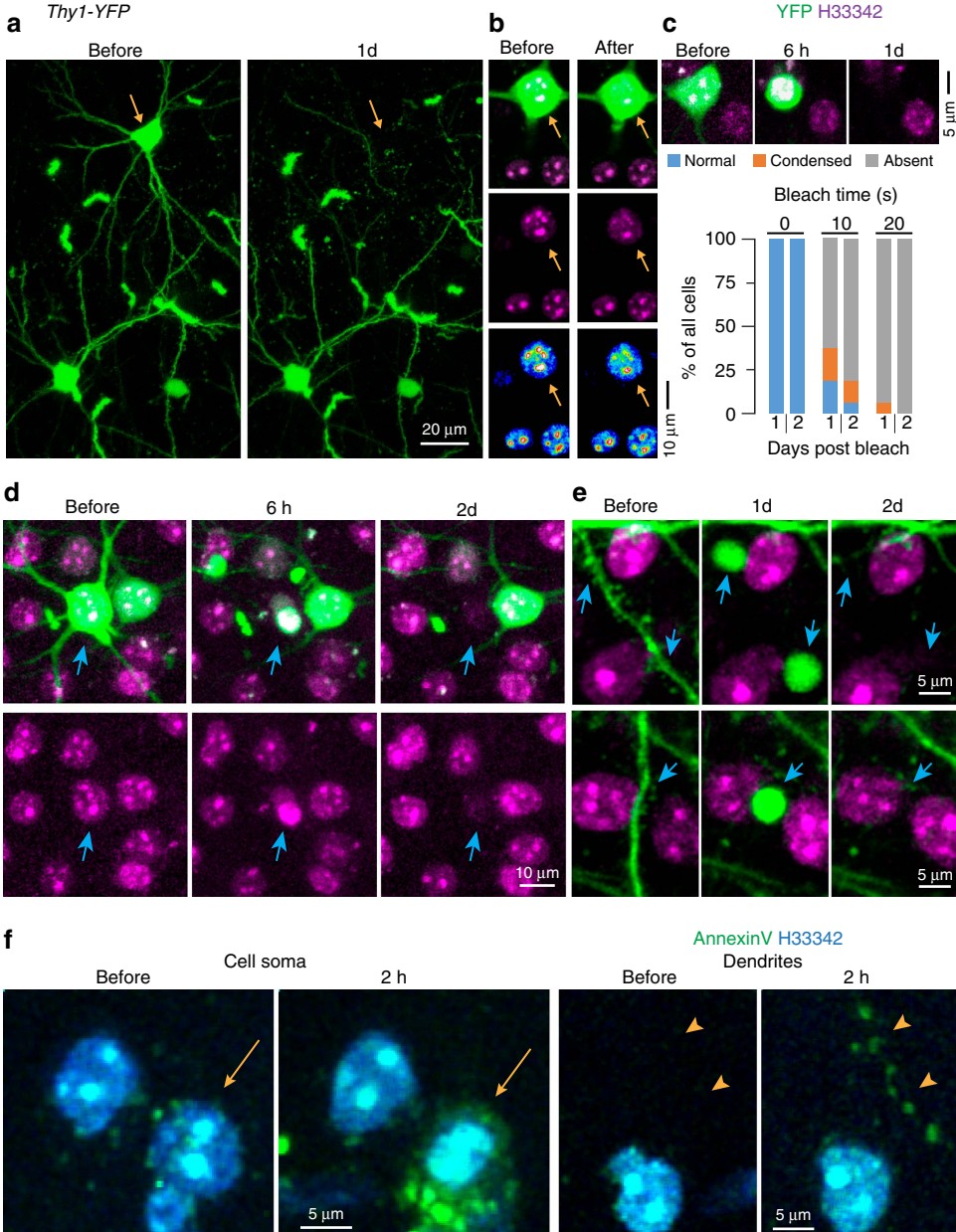

**Figure 4 | Nuclear pyknosis and formation of apoptotic bodies in targeted neurons.** (**a**) *In vivo* images captured from the cortex of a *Thy1-YFP* mouse showing single neuron ablation over 1 day (yellow arrows). (**b**) Photobleaching of H33342 causes no immediate changes in YFP labelling or cell soma integrity (arrow indicates bleached cell). (**c**) Laser scan time-dependent cell death induction 1 and 2 days after photobleaching, $n = 24$ cells, 0 s control; 16 cells, 10 s scan; and 16 cells, 20 s scan, from 4 mice. (**d**) Time-lapse sequence showing nuclear pyknosis and degeneration of cell soma and dendrites (blue arrows) after 2Phatal. Top panel shows combined YFP and H33342 signals, whereas bottom panels show only H33342 fluorescence. (**e**) Single dendrites imaged over 2 days showing their transformation into apoptotic bodies (arrows) during apoptosis of a single neuron after 2Phatal. (**f**) *In vivo* images showing a single neuron becoming labeled with Annexin V at the cell soma (arrows) and along single dendrites (arrowheads) $n = 15$ cells from 4 mice.

distribution, suggesting an alteration in permeability at the nuclear envelope[28] (Fig. 5d and Supplementary Movie 5). Quantification before and after 2Phatal induction revealed a significant decrease in calcium spike events ($n = 11$ cells from 4 mice) compared with normal adjacent cells ($n = 14$ cells from 4 mice) at both 2 h ($P = 0.0001$) and 6 h ($P = 0.0213$), and in the overall fluorescence fluctuations ($P = 0.0102$ at 2 h, $P = 0.0267$ at 6 h) (significance determined by multiple *t*-tests with Holm–Sidak correction) (Fig. 5e,f). These results demonstrate *in vivo* a sustained alteration in calcium homeostasis at early stages of the apoptotic process.

Feed-forward signals link endoplasmic reticulum calcium release and mitochondrial function during early stages of apoptosis[26,29]. To precisely monitor mitochondrial dynamics at different stages of the apoptotic process, *in utero* electroporation was used to fluorescently tag mitochondria in Layer II neurons (Fig. 6a). *In vivo* time-lapse imaging of cells subjected to 2Phatal revealed a progression of mitochondrial fission and fragmentation in cell bodies and along individual dendrites starting at 3 h after photobleaching (Fig. 6a). Quantification revealed a significant decrease in mitochondria size and in their spread along dendrite branches ($n = 15$ dendrites from 3 mice)

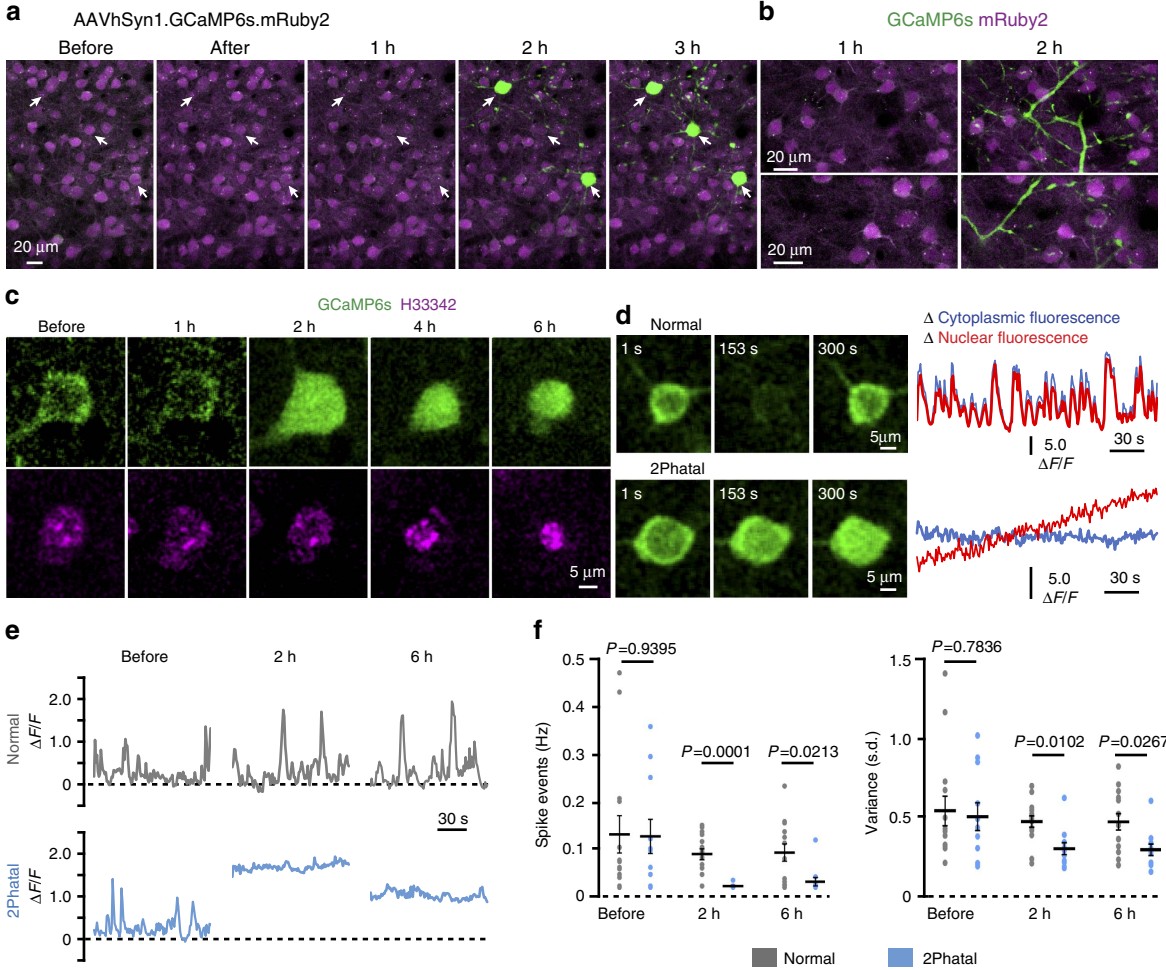

**Figure 5 | Temporal course of intracellular calcium changes during apoptosis. (a)** *In vivo* time-lapse images of AAV.GCaMP6s.mRuby2-labelled cells in the cortex showing sustained increases in GCaMP6s fluorescence 2–3 h after photobleaching of three cells targeted for 2Phatal (arrows). **(b)** Images captured of single dendrites demonstrating sustained calcium increases in cell processes coinciding with calcium increases in soma. **(c)** *In vivo* time-lapse sequence showing a single neuron targeted for 2Phatal with increases in GCaMP6s signal coinciding with initiation of nuclear condensation. **(d)** Time-lapse sequence showing single images and representative fluorescence intensity traces of nuclear and cytoplasmic GCaMP6s calcium signals in normal and photobleached cells, suggesting alteration in permeability at the nuclear envelope during apoptosis. **(e)** Spontaneous GCaMP6s fluorescence intensity traces from awake head fixed mice at the indicated time points in normal control cells or in cells subjected to photobleaching. **(f)** Left graph showing quantification of the spontaneous frequency of spike events before, 2 and 6 h after; significance determined by multiple *t*-tests with Holm–Sidak correction, before $P = 0.9395$, $t = 0.0767$, df $= 23$; 2 h $P = 0.000124$, $t = 5.05$, df $= 23$; 6 h $P = 0.0213$, $t = 2.7778$, df $= 23$, as indicated. Right graph shows quantification of the variance (s.d.) within a single 120 s imaging trial before, 2 and 6 h after 2Phatal induction; significance determined by multiple t-tests with Holm–Sidak correction before $P = 0.7836$, $t = 0.2779$, df $= 23$; 2 h $P = 0.0102$, $t = 3.263$, df $= 23$; 6 h, $P = 0.0267$, $t = 2.678$, df $= 23$, as indicated. Normal $n = 14$ cells from 4 mice, 2Phatal $n = 11$ cells from 4 mice (df $=$ degrees of freedom).

compared with normal adjacent cells ($n = 14$ from 3 mice) at both 3 h ($P < 0.0001$) and 6 h ($P < 0.0001$) after initial photobleaching (significance determined by multiple *t*-tests with Holm–Sidak correction) (Fig. 6b and Supplementary Fig. 5). These data demonstrate for the first time in the live mouse brain a temporally coordinated sequence of apoptosis-associated calcium overload and mitochondrial fission. Thus, we conclusively show that 2Phatal can be used to precisely dissect defined subcellular processes at high spatio-temporal resolution during DNA damage-induced apoptosis *in vivo*.

**Fast-spiking interneuron ablation disrupts neuronal activity.** We next determined whether 2Phatal could be used to test the functional consequences of the loss of distinct neuronal subtypes to local neuronal circuit function *in vivo*. Layer II/III parvalbumin (PV)-expressing fast-spiking interneurons were targeted

in transgenic *PV*cre:Tomato reporter mice with viral mediated GCaMP6 pan neuronal labelling (Fig. 7). PV+ interneuron processes in these mice target the perisomatic region of neighboring GCaMP6f-labelled pyramidal cells (Fig. 7b); thus, we hypothesized that 2Phatal of the local PV+ cell population would alter the spontaneous activity of remaining neurons. Somatic calcium levels from GCaMP6f-labelled cells were imaged in awake head-fixed mice over multiple trials before and 1 day after 2Phatal of PV+ interneurons (Fig. 7c,d). 2Phatal applied to a patch of 8–11 PV+ neurons resulted in a significant increase in spike events specifically in cells adjacent to ablated cells ($P = 0.019$, $n = 4$ mice) but not in control regions in the same mouse where no cells were ablated ($P = 0.234$, $n = 4$ mice) (significance determined by paired *t*-tests) (Fig. 7e). Thus, 2Phatal can be used to study the discrete functional consequences of targeted cell removal on the local cortical circuitry.

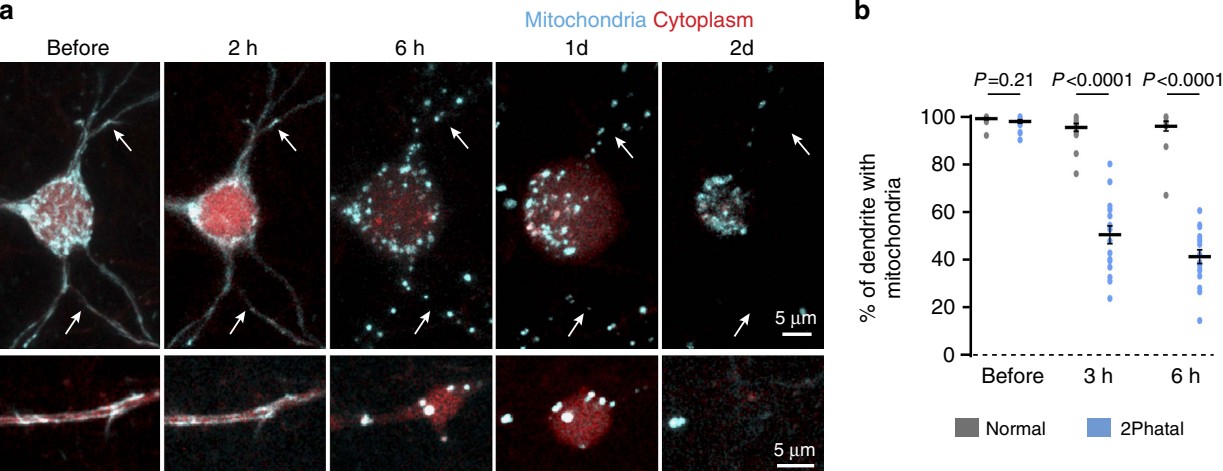

**Figure 6 | Mitochondrial fission during apoptosis.** (**a**) *In vivo* time-lapse sequence showing mitochondrial fission occurring in cell soma and dendrites between 2 and 6 h after photobleaching. Arrows indicate the location of the same dendritic branch for each time point. (**b**) Quantification of mitochondria fission in normal vs photobleached cells based on changes in the percentage of single stretches of dendrite filled with mitochondrial fluorescence labelling; significance determined by multiple *t*-tests with Holm–Sidak correction, before $P = 0.21$, $t = 1.286$, df = 31; 2 h, $P < 0.0001$, $t = 10.76$, df = 31; 6 h, $P < 0.0001$, $t = 15.35$, df = 31, as indicated. Normal $n = 14$ dendrites from 3 mice, 2Phatal $n = 15$ dendrites from 3 mice (df = degrees of freedom).

**Cell-type-specific progression of apoptosis**. Next, we determined whether we could investigate apoptotic mechanisms in non-neuronal cell types. We first tested the effectiveness of 2Phatal to induce astrocyte apoptosis. H33342-labeled astrocytes were identified by concomitant SR101 labelling or in transgenic *Aldh1L1*cre:Z/EG cytoplasmic GFP reporter mice (Fig. 8). Similar to neurons, brief bleaching of H33342 within astrocytes induced nuclear pyknosis and formation of apoptotic bodies (Fig. 8b and Supplementary Fig. 6) leading to cell death that was also proportional to laser scan time ($n = 18$–20 cells per condition, from 4 mice for SR101 labelling). Astrocyte apoptosis initiation was not dependent on the method of astrocytic labeling as GFP-only-labelled astrocytes and SR101-labelled astrocytes initiated apoptosis at similar time points (Supplementary Fig. 6). Interestingly, unlike neurons that consistently died within 1–2 days after photobleaching (Fig. 4c), astrocyte apoptosis initiation was delayed until 4–5 days (Fig. 8b), despite both their initial brighter nuclear labelling and subsequent greater degree of photobleaching (Fig. 1c and Supplementary Fig. 7). This previously unknown difference in the initiation of apoptosis between neurons and astrocytes suggested fundamental cell intrinsic differences in the response to DNA damage between these cell types.

The protracted initiation of cell death in astrocytes provided a unique time window for investigation into subcellular response mechanisms occurring during this 3–4-day period preceding terminal apoptosis. We first tested at what stage mitochondrial fission events occurred in astrocytes by using cre-dependent GFP mitochondria reporter mice (ACTINcre:PhAM; see Methods) (Fig. 8c). We found that mitochondrial fission in astrocytes occurred 3–4 days after *2Phatal* induction, consistent with our finding that the initiation of nuclear condensation was markedly delayed in astrocytes (3–4 days) compared with neurons (2–4 h) (Supplementary Fig. 7). In contrast to the late astrocytic induction of nuclear pyknosis and mitochondrial fission, live imaging of *Aldh1L1*-EGFP bacTRAP transgenic mice, in which the ribosomal subunit L10a is tagged with enhanced GFP (EGFP)[30], suggested that ribosomes were rapidly disassembled 1 day after 2Phatal induction, well before any other cellular changes associated with apoptosis (Fig. 8e–g, Supplementary Fig. 6 and Supplementary Movie 6, $n = 12$ cells for each condition from 3 mice). Thus, ribosomal disassembly and degradation

may constitute one of the earliest mechanisms during DNA damage-mediated apoptotic death.

**Ablation of NG2 glia, pericytes and zebrafish hair cells**. We next tested whether 2Phatal was effective at killing of other cell types in the adult central nervous system. NG2 glia (also known as oligodendrocyte progenitor cells or polydendrocytes[31]) and vascular pericytes were identified and imaged *in vivo* (constitutive and inducible *NG2*cre:mT/mG, *NG2*cre:ZEG and *NG2*creER:mT/mG transgenic mice were used, see Methods)[32] (Fig. 9a). For *NG2*creER:mT/mG mice, low-dose tamoxifen injections were used to label single cells[33]. *In vivo* time-lapse imaging of NG2 glia subjected to 2Phatal resulted in nuclear pyknosis and formation of apoptotic bodies along single processes (Fig. 9b,c and Supplementary Fig. 8) leading to apoptotic death over 24 h. Neighbouring non-ablated NG2 glia processes invaded the territory of the ablated cell over the following days (Supplementary Fig. 8). Similarly, *in vivo* imaging of vascular pericytes subjected to 2Phatal revealed apoptotic cell death over 2–3 days (Fig. 9). In addition to testing the efficiency of 2Phatal for cell ablation, mitochondria were fluorescently tagged in NG2 glia and vascular mural cells, and imaged *in vivo* during the apoptotic process (Fig. 9e). Consistent with our findings in neurons and astrocytes, both NG2 glia and pericytes displayed mitochondrial fission events coinciding with nuclear condensation (Fig. 9f,g). Thus, 2Phatal provides a simple, powerful tool for exploration of apoptotic mechanisms and functional roles played by these understudied central nervous system cell populations.

Finally, we tested whether 2Phatal could also be used in zebrafish. H33342 was used to label neuromast hair cells in the lateral line, a mechanosensory cell type similar to the mammalian inner ear hair cells that is used by zebrafish for navigation (Fig. 10a,b). Using *2Phatal*, we induced death of single hair cells labelled in a *Prox1*-RFP reporter transgenic line (Fig. 10c–e). Time-lapse imaging of single cells revealed consistent nuclear pyknosis followed by what appeared to be extrusion of the condensed nuclei from the local cluster of cells (Fig. 10e and Supplementary Movie 7). Single cells followed a stereotyped condensation and extrusion behaviour ranging from 1 to 6 h

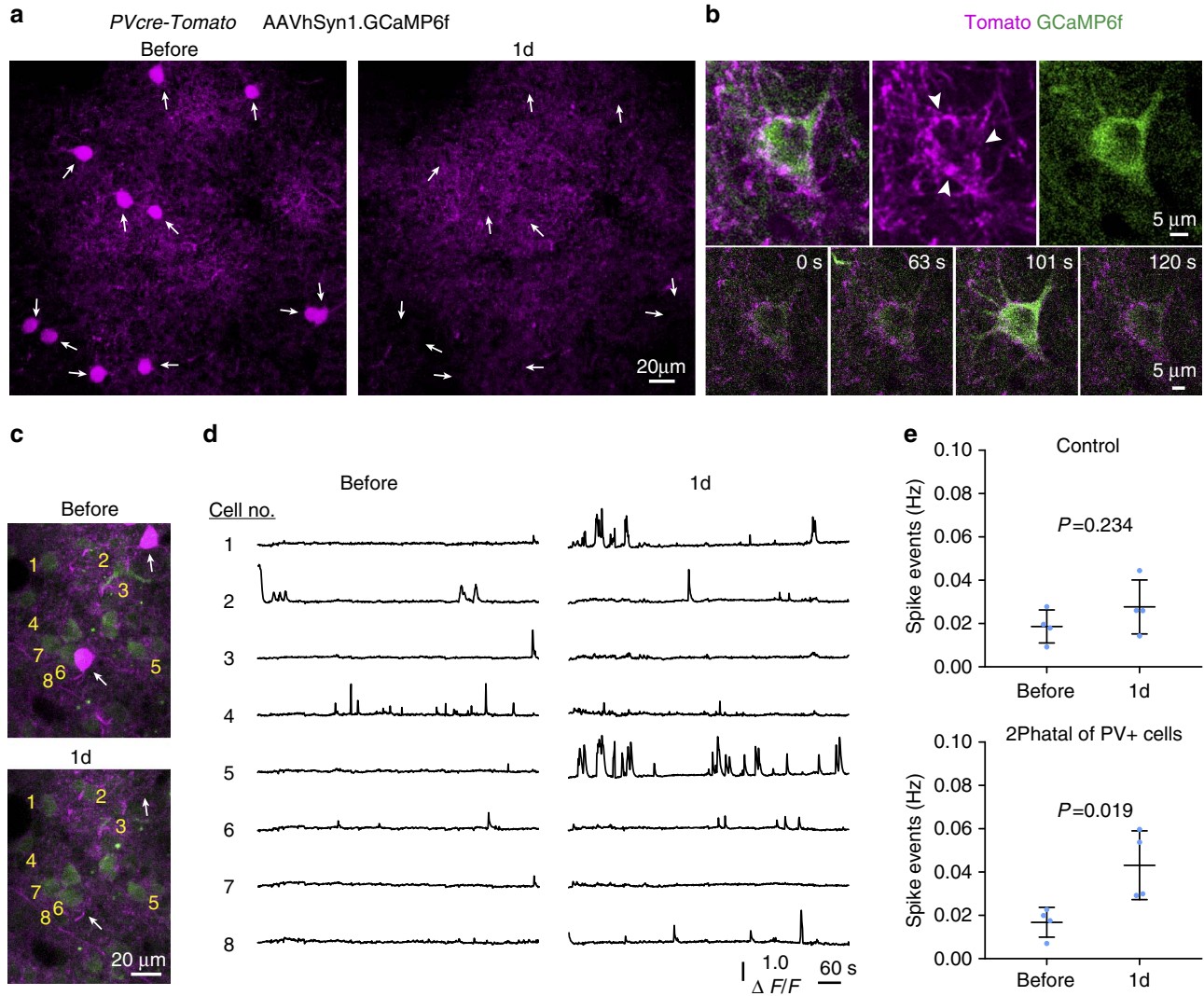

**Figure 7 | Disrupted neuronal activity after removal of fast-spiking interneurons.** (**a**) *In vivo* images showing 2Phatal ablation of 11 fast-spiking cortical interneurons (arrows) in a transgenic mouse with tdTomato labelling specifically in PV-expressing neurons (*PV*cre-Tomato). (**b**) High-resolution *in vivo* image (top) and time-lapse sequence (bottom) of a single GCaMP6f AAV viral-labelled neuron with *PV*cre-Tomato-labelled presynaptic terminals surrounding the cell soma (arrowheads). (**c**) *In vivo* images showing 2Phatal of single PV + interneurons (arrows) and location of adjacent GCaMP6f–labelled neurons that were imaged in awake head-fixed mice before and after 2Phatal. (**d**) Spontaneous GCaMP6s fluorescence intensity traces from awake head-fixed mice taken from the numbered cells indicated in (**c**) before and 1 day after 2Phatal of PV + interneurons. (**e**) The spontaneous frequency of calcium transients in control regions with no cells ablated or after 2Phatal of 8–11 PV + interneurons within the local area. Significance determined by paired *t*-tests, control $P = 0.234$, $t = 1.485$, df = 3, 2Phatal, $P = 0.019$, $t = 4.677$, df = 3, as indicated, $n = 4$ mice (62 cells in control group, 67 cells in 2Phatal group) (df = degrees of freedom).

($n = 9$ cells from 3 fish) after initial photobleaching (Fig. 10f), remarkably similar to the time course of neuronal apoptosis in the mouse cerebral cortex. Thus, 2Phatal can be used to study cell death mechanisms and functional consequences of single-cell removal in both mice and zebrafish, and likely in many other species.

## Discussion
We have developed a powerful platform for spatiotemporally targeted single cell ablation in the live animal. The following features set 2Phatal apart from previous cell ablation methods. First, rapid nucleic acid dye photobleaching with a femtosecond-pulsed two-photon laser results in targeted apoptosis evidenced by stereotyped nuclear pyknosis, formation of apoptotic bodies, phosphatidylserine membrane exposure,

calcium overload and mitochondrial fission. Second, it does not induce acute microglial activation or collateral damage to neighbouring cells or immediately adjacent synapses. Third, at the concentrations we used, nucleic acid dyes are non-toxic, can be used to image nuclear dynamics and are cleared from the cells over short times. Fourth, this method can be relatively high throughput due to the short bleaching times and can be adopted for multiple cell types and animal species with conventional two-photon systems and without the requirement of time-consuming and often not feasible molecular manipulations. Thus, 2Phatal allows precise dissection of cellular and molecular mechanisms of DNA damaged-induced apoptosis, cell debris clearance and functional consequences of single-cell loss for the first time in the live animal.

Similar photochemical methodologies have been previously used mostly for single molecule inactivation via light induced

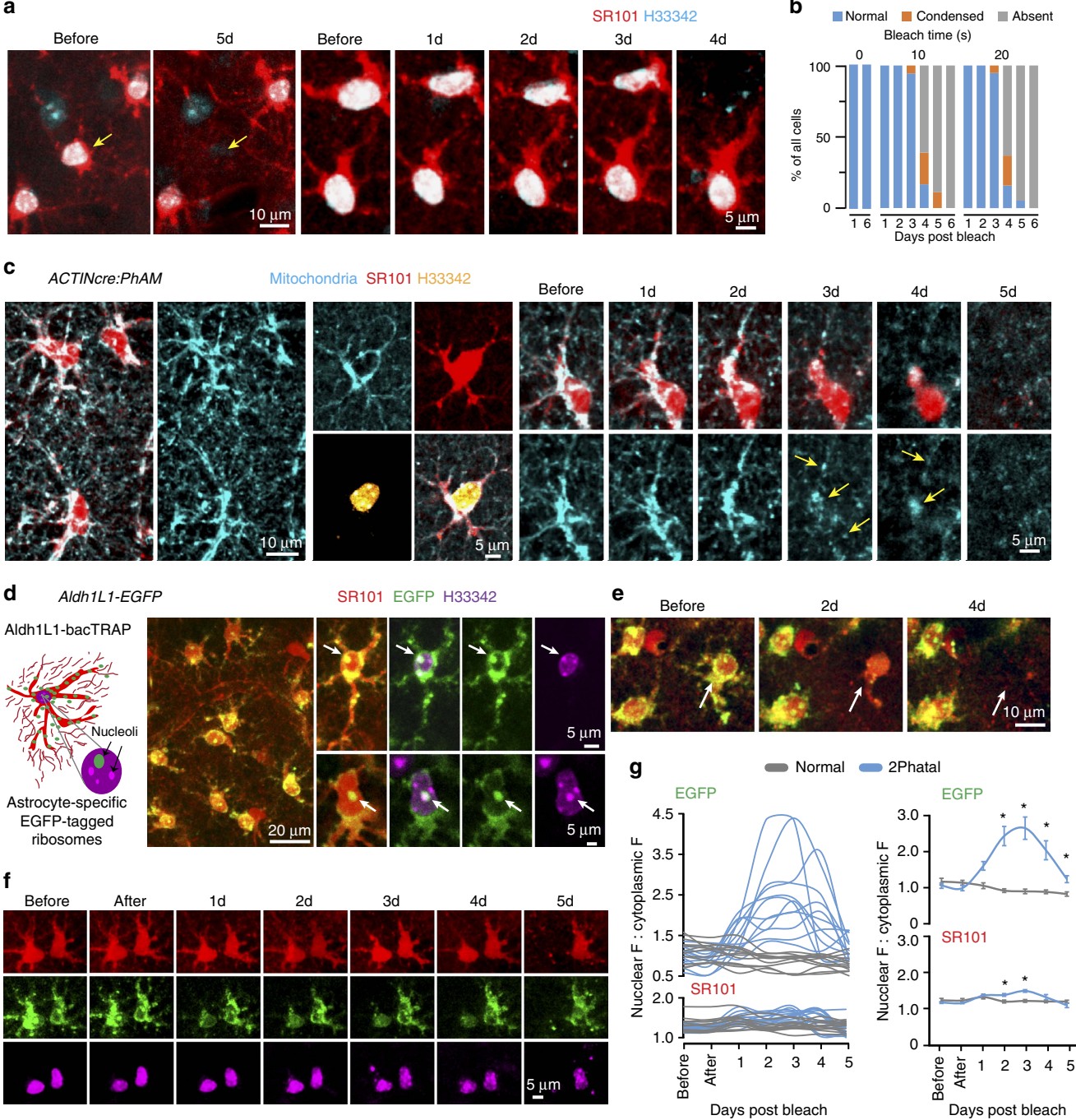

**Figure 8 | Delayed apoptosis initiation and early ribosomal disassembly in astrocytes.** (**a**) *In vivo* time-lapse images of SR101-labelled astrocyte (yellow arrows) dying after photobleaching. (**b**) Laser scan time-dependent astrocyte-specific cell death induction 1–6 days after 2Phatal photobleaching, $n = 20$ cells, 0 s control; 18 cells, 10 s scan; and 19 cells, 20 s scan; from 4 mice. (**c**) SR101-labelled astrocytes showing mitochondrial fission (yellow arrows) occurring 3–4 days after initial photo bleaching. (**d**) Schematic showing EGFP targeting to astrocyte ribosomes in *Aldh1L1*-EGFP bacTRAP transgenic mice and *in vivo* images of SR101-labelled astrocytes with EGFP-labelled ribosomes in the cytoplasm and nucleoli (arrows). (**e**) Imaging of a single astrocyte (arrow) showing maintenance of SR101 labelling 2 days after photobleaching, whereas EGFP-L10a fluorescence labelling is markedly reduced. (**f**) *In vivo* time-lapse sequence showing loss of EGFP-L10a label in a single astrocyte 1 day after photobleaching, whereas SR101 labelling remains until the day of nuclear condensation and cell death progression. (**g**) Quantification of the nuclear and cytoplasmic fluorescence intensity ratios for EGFP-L10a and SR101 at sequential days after photobleaching showing both single cell traces and grouped data with mean ± s.e.m., two-way analysis of variance with Bonferroni *post hoc* comparing EGFP-L10a in normal vs 2Phatal for days 1–5 after bleaching $n = 12$ cells for each condition from 3 mice. GFP: before; $P > 0.9999$, $t = 0.5440$, df $= 154$; after: $P > 0.9999$, $t = 0.7102$, df $= 154$; 1d $P = 0.0615$, $t = 2.654$, df $= 154$; 2d $P < 0.0001$, $t = 7.296$, df $= 154$; 3d $P < 0.0001$, $t = 8.450$, df $= 154$; 4d $P < 0.0001$, $t = 5.554$, df $= 154$; 5d $P = 0.3392$, $t = 1.989$, df $= 154$. SR101: before $P > 0.9999$, $t = 0.9644$, df $= 154$; after $P > 0.9999$, $t = 1.079$, df $= 154$; 1d $P > 0.9999$, $t = 0.6476$, df $= 154$; 2d $P = 0.0338$, $t = 2.860$, df $= 154$; 3d $P = 0.0002$, $t = 4.346$, df $= 154$; 4d $P = 0.4797$, $t = 1.834$, df $= 154$; 5d $P = 0.6376$, $t = 1.700$, df $= 154$ (df = degrees of freedom).

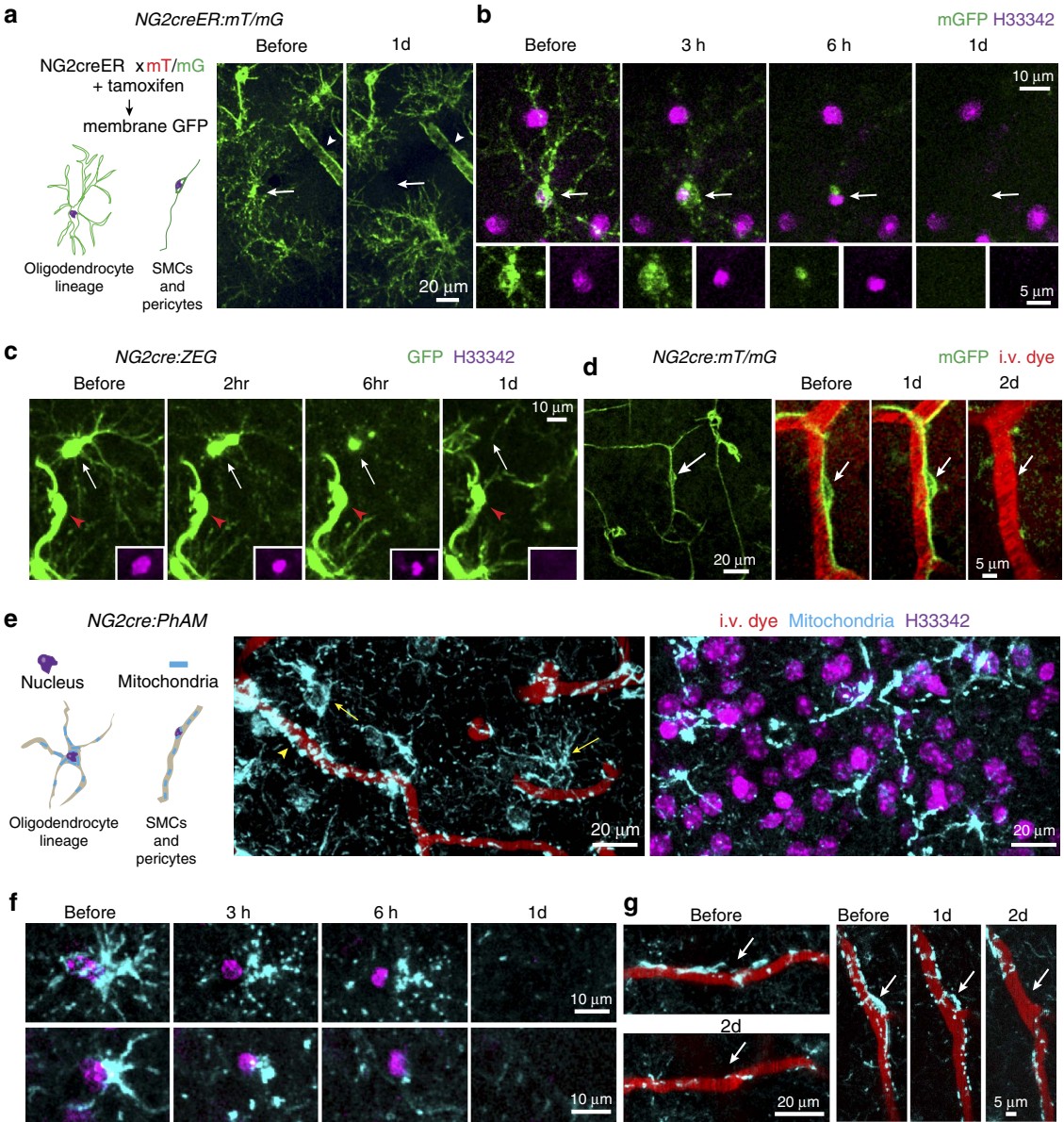

**Figure 9 | 2Phatal of NG2 glia and vascular pericytes.** (**a**) Schematic showing membrane anchored GFP targeted to oligodendrocyte lineage cells and vascular mural cells after tamoxifen administration and *in vivo* images showing ablation of a single NG2 glial cell (arrow) with a smooth muscle cell indicated (arrowhead). (**b,c**) *In vivo* time-lapse sequences showing nuclear pyknosis and formation of apoptotic bodies after photobleaching in single NG2 glia (arrows). Vascular mural cells are indicated (red arrowheads). (**d**) Ablation of a single vascular pericyte (arrow) at the time points indicated. (**e**) Schematic showing mitochondrial labelling in oligodendrocyte lineage cells and vascular mural cells in *NG2*cre:PhAM transgenic mice. (**f**) *In vivo* time lapse sequence showing nuclear pyknosis and mitochondrial fission in single NG2 glia over 6 h. (**g**) *In vivo* time-lapse images showing pericyte (arrows) mitochondrial fission and cell removal over 2 days.

local production of ROS (CALI)[4,7,18,34]. Chromophores used in past studies range from exogenously applied dyes such as Malachite green[4] and Fluorescein[7] to genetically encoded fluorescent proteins such as EGFP[6], KillerRed[8], miniSOG[9,10] and SuperNova[35] targeted to the cell or molecule of interest. CALI has mainly been used in cell culture systems and, in some cases, optically accessible transparent animal models such as *Caenorhabditis elegans*, *Drosophila melanogaster* and zebrafish[4,5,7,10,36,37]. Importantly, previous approaches have not been successfully used in the live mammalian brain due to difficulties in targeted cell labeling and prohibitively long illumination protocols (on the order of minutes to hours, in contrast to seconds with 2Phatal), with significant off target damage associated with the non-focal illumination methods.

The focal illumination properties of femtosecond pulsed lasers could overcome these issues and two-photon inactivation of GFP-labelled proteins has indeed been demonstrated[19]. However, *in vivo* two-photon CALI with fluorescent proteins has not been established and, based on our data, its extension to cell ablation *in vivo* is likely to be limited (Supplementary Fig. 1). 2Phatal overcomes these issues by using a femtosecond pulse laser to cause highly focal and very brief photobleaching (5–10 s) of a nuclear-targeted photosensitizer. This provides a tunable and reproducible system for single cell *in vivo* induction of apoptosis without the off target effects seen with other ablation methods.

In addition to photochemical methods, other approaches for cell ablation have been developed based on transgenic expression of toxin receptors or apoptotic pathways[15–17]. However,

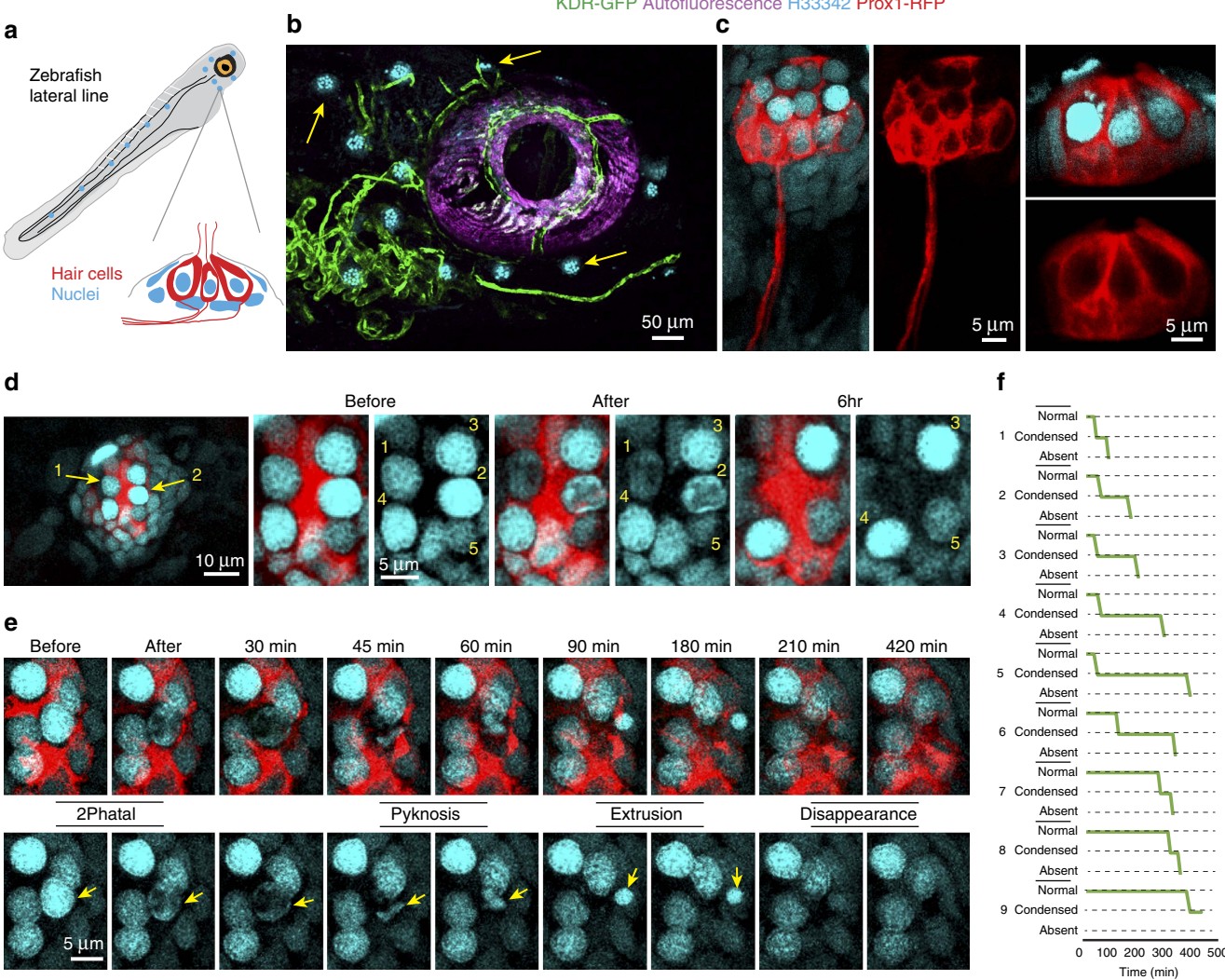

**Figure 10 | Ablation of neuromast hair cells in the zebrafish lateral line.** (**a**) Schematic depicting zebrafish neuromast hair cells of the lateral line labeled with H33342. (**b**) Low magnification live image of an anesthetized *KDR*-GFP transgenic zebrafish with the neuromasts labelled with H33342 (yellow arrows) and GFP-labelled vessels shown in green. (**c**) High-resolution live images of single neuromasts with hair cells labelled in the *Prox1*-RFP transgenic line and cell nuclei labelled with H33342. (**d**) Two hair cells targeted for 2Phatal (yellow arrows) and imaged before, immediately after and at 6 h showing disappearance of the targeted cells. Numbers designate single cells imaged over time. (**e**) Time-lapse sequence showing nuclear pyknosis, extrusion of the condensed nucleus and disappearance of the targeted cell (yellow arrows). (**f**) Traces showing the time course of the apoptotic progression for nine cells targeted for 2Phatal from three separate fish.

genetically targeting specific cell populations can be prohibitive due to lack of cell type specificity and inability to exclusively deliver genetic modifiers to defined cell populations in the live brain or other organs. This is particularly true for populations of cells with overlapping molecular expression in which genetically encoded systems could result in off target cell ablation. Furthermore, unlike 2Phatal, these approaches do not permit experimental targeting of single cells for ablation. For a direct example, we used 2Phatal for specific ablation of NG2 glia and vascular pericytes, which is not currently possible using genetically encoded Diphtheria toxin receptor due to a lack of exclusive molecular identifiers for each cell population. Thus, as one example, this approach provides a means to investigate in the intact brain the roles of NG2 glia separately from the function of pericytes in vascular homeostasis.

To date, attempts at spatiotemporal targeted ablation in zebrafish and mammalian systems have been limited to long ultraviolet illumination or two-photon thermal ablation[11,13,14,21–24]. However, ultraviolet illumination causes membrane damage and even mild thermal ablation protocols cause spilling of cellular contents, which induce rapid microglial activation (Fig. 3 and Supplementary Movie 2)[25,38]. This prohibits studies of apoptotic mechanisms or fine structural and functional investigation of adjacent non-ablated cells. 2Phatal overcomes these limitations by initiating apoptosis at the single-cell level with low laser intensity and without acute microglial activation or damage to adjacent cells.

Very little is known about dynamic cellular and molecular events occurring during single-cell apoptosis in intact living organisms. 2Phatal now allows studies of these apoptotic mechanisms in multiple organs and species. We used 2Phatal to show distinct phases of single-cell apoptosis revealing calcium overload and mitochondrial fission events in neuronal cell bodies and processes. Furthermore, our preliminary findings suggest a previously unrecognized process of ribosomal disassembly at very early stages of apoptosis well before calcium overload, nuclear condensation and mitochondrial changes. It is possible that these findings represent breakdown in polysome structure[39] and

indicate decreased protein translation early during the commitment phase of the apoptosis cascade[40–42]. In addition, application of 2Phatal to multiple cell types in the nervous system revealed intriguing temporal differences in the initiation phases of the apoptotic cascade, which probably represent unique cell-type-specific DNA damage responses or ROS buffering mechanisms[43,44]. Thus, 2Phatal now provides a means to investigate these phenomena in a physiological *in vivo* system.

Once apoptosis has occurred, phagocytic mechanisms of cell clearance are critical for maintaining tissue homeostasis[24,45,46]. *2Phatal* also provides a new platform for investigating such mechanisms of clearance of apoptotic cells in numerous genetically modified organisms to dissect molecular and cellular pathways involved in physiological and pathological phagocytosis by microglia and other cells. For example, our intriguing observation that dying zebrafish hair cells may be extruded from the neuromast is consistent with recent observations of cell extrusion events in developing *Xenopus laevis* brain[47]. It is likely to be that such mechanisms are found in multiple developing organ systems and 2Phatal provides a robust system for studying them.

Finally, once cells are removed, there are likely to be robust tissue remodelling changes and circuit-based disturbances. 2Phatal of a focal population of fast spiking interneurons resulted in a local increase in spontaneous activity in neighboring cells. These findings are consistent with data acquired with optogenetic silencing approaches[48] however additional experiments are necessary to discern the cellular mechanisms behind this finding. Importantly, 2Phatal now provides a system for studying these and other consequences of single cell removal in live animals. Combined with functional probes of calcium and voltage, removal of distinct neurons or astrocytes could be used for targeted single-cell disruption and interrogation of neural circuits.

## Methods

**Animals.** All animal procedures were approved by the Institutional Animal Care and Use Committee. Both male and female mice aged P30–P150 housed in a 12/12 h light/dark cycle housed 2–5 per cage were used in these studies. No animals were excluded from analysis. The following transgenic mouse lines were used for visualization of defined cell populations in the live brain: *Thy1*-YFPh[49] (Jackson Labs 003782), *Thy1*-GFPm (Jackson Labs 007788), *CX3CR1*-GFP[50] (Jackson Labs 005582), *PV*cre[51] (Jackson Labs 008069), *Aldh1L1*cre[52] (Jackson Labs 023748), *Aldh1L1*-bacTRAP-EGFP[30] (provided by Dr Flora Vacarino, Yale University), *NG2*cre[53] (Jackson Labs 008533) also used as ACTINcre based on cre recombination in male germ cells[54], *NG2*creER[55] (Jackson Labs 008538), *SST*cre[56] (Jackson Labs 013044), nT/nG[57] (Jackson Labs 023035), tdTomato Ai9 (ref. 58) (Jackson Labs 007909), PhAM[59] (Jackson Labs 018385), Z/EG[60] (Jackson Labs 003920) and mT/mG[61] (Jackson Labs 003920). Zebrafish were imaged from 48 to 96 h post fertilization. The following transgenic zebrafish lines were used: Tg(*kdrl*:EGFP) (*KDR*-GFP)[62] and TgBAC(*prox1a*:KALTA4,4xUAS-ADV.E1b:TagRFP) (*Prox1*-RFP)[63].

**Surgery and *in vivo* imaging.** For all mouse experiments, cranial windows were used. Briefly animals were anaesthetized via intraperitoneal injections of 100 mg kg$^{-1}$ ketamine and 10 mg kg$^{-1}$ xylazine or via inhaled isoflurane. A region of the skull (3 × 3 mm) was gently removed with a high speed drill and the underlying dura was removed. A small 0 glass coverslip was placed over the skull to allow long term optical access for *in vivo* imaging. In some cases, cerebral vessels were visualized by intravenous injection of 70,000 MW Texas Red dextran (ThermoFisher catalogue number D1830). Repeated SR101 labelling of astrocytes and oligodendrocytes was performed via intravenous injections[64]. In some cases, Alexa-488 Annexin V (ThermoFisher catalogue number A13201, 1:50 dilution) was applied topically to the cortical surface for one hour and washed for 10 min to detect phosphatidylserine on apoptotic cells.

*In vivo* images were acquired using a two-photon microscope (Prairie Technologies) equipped with a mode-locked MaiTai two-photon laser (Spectra Physics) and × 20 water immersion objective (Zeiss 1.0 numerical aperture). For all experiments except GCaMP6s calcium imaging, animals were anaesthetized with ketamine/xylazine or inhaled isoflurane. For GCaMP6s imaging, time-lapse images were acquired in awake head-fixed mice as described[33] The two-photon laser was tuned to the following wavelengths for optimal excitation of particular fluorophores: 775 nm for H33342 and Tomato; 900 nm for GFP, yellow fluorescent protein (YFP), mGFP, mTomato, nTomato, GCaMP6s, SR101, Annexin V, mRuby2 and Texas Red dextran; and 1,030 nm for Tomato. In some cases, to faithfully detect and display co-localization between 775 nm excited H33342 and fluorescent proteins only excited by 900 nm, sequential multi-wavelength imaging was used. Images were displayed as an overlay to visualize both nuclear dye and fluorescent proteins.

**2Phatal.** To induce photobleaching and apoptotic cell death, nuclei were labelled *in vivo* using Hoechst 33342 (ThermoFisher catalogue number H5370). The dye was either applied topically (0.04–0.01 mg ml$^{-1}$ diluted in PBS) to the cortical surface for 10 min and then thoroughly washed with PBS or injected intravenously (50 μl volume at 10 mg ml$^{-1}$) (Supplementary Fig. 1). Labelling was evident within 2 h and remained for ~7–10 days (Supplementary Fig. 1). To induce 2Phatal, single ROIs (20 × 20 pixels, 8 × 8 μm) were selected and centred on defined nuclei. Laser wavelength was set to 775 nm and pixel dwell time was set to 100 μs. Photobleaching was achieved by 5–20 s laser scanning per cell as indicated in the text with laser intensities varying between 21.3 and 45.4 mW as indicated, measured at the objective-sample interface with a power meter. After initial optimization of laser scan time and intensity (Figs 1–4), a 10 s laser scan time per cell with defined 8 × 8 μm ROI was used for all subsequent experiments.

**Viral infection and *in utero* electroporation.** The following viruses were used to label neurons with GCaMP6 as indicated. Viruses were obtained from Penn Vector Core and based on Addgene plasmid 50942: AAV1.hSyn1.mRuby2.GSG.P2A.GCaMP6s.WPRE.SV40 (lot CS0493) with a titre of 3.69e13 GC ml$^{-1}$ (Fig. 5) or AAV9.Syn.GCaMP6f.WPRE.SV40 (lot CS1001) with a titre of 7.648e13 GC ml$^{-1}$ (Fig. 7). Viruses (diluted 1:10 in PBS) were delivered via injection into the subarachnoid space resulting in widespread labeling of cerebral Layer II/III neurons or superficial astrocytes without direct injection in the cortex. The following plasmids were used for *in utero* electroporation to label mitochondria: tdTomato-mito-7 (ref. 65) (Addgene 58115). *In utero* electroporation was conducted at embryonic day 15.5 to label Layer II neurons[66]. Briefly, timed pregnant mice were anaesthetized with a mixture of ketamine and xylazine. The abdominal region was shaved, sterilized and a 3 cm midline incision was made in the skin and abdominal muscle. The uterine horns were exposed and the lateral ventricle of each embryo was pressure injected (Picospritzer II, General Valve) with plasmid DNA (~0.5 μl volume per embryo) at a concentration of 1 μg μl$^{-1}$ followed by electroporation with tweezertrodes (50 V, 4–50 ms pulses with 1 s pulse interval, BTX Harvard Apparatus) to target Layer II cortical neurons. The embryos were placed back in the mother, and the muscle and skin were sutured. Electroporated pups were aged to postnatal day 30 and a craniotomy was performed over the transfected hemisphere to carry out 2Phatal experiments as described above.

**Zebrafish dye labelling and imaging.** Zebrafish aged 49–96 h post fertilization were labelled with H33342 applied in the water at a concentration of 2.5 μg ml$^{-1}$ for 30 min. Fish were then anaesthetized with tricaine methane-sulfonate (Sigma Aldrich) and embedded in 1% low melting point agarose during imaging. Time-lapse images were acquired on a Leica SP5 confocal microscope with a × 20 water immersion objective (Leica 1.0 numerical aperture) at intervals of 10–15 min for 4–8 h. 2Phatal photobleaching of single neuromast hair cells was carried out on the two-photon microscope as described above. Time-lapse movies were aligned, analysed and quantified in ImageJ to characterize nuclear condensation, extrusion and disappearance as indicated.

**Quantification and data analysis.** For quantification of cell death after 2Phatal photobleaching (Figs 1,4 and 8), cell nuclei were characterized as normal, condensed or absent as indicated in Fig. 1g at the days indicated in the text and figures. Quantification was performed blind to laser intensity and scan time for each experiment. For quantification of acute microglia response to thermal ablation or 2Phatal (Fig. 2c), time-lapse images were acquired once per minute for 15 min before 2Phatal or thermal ablation and then once per minute for 15 min following. Changes in CX3CR1-GFP fluorescence intensity indicated as $\Delta F/F$, where $F$ is initial fluorescence intensity within a circular ROI (radius 50 μm) centred on the ablated cell were measured for each time point and increase in the fluorescence intensity was used as an indication of microglia activation (or lack thereof) towards the targeted cell as shown in Fig. 2. For quantification of axonal bouton plasticity (Fig. 3b), all boutons within a 20 × 20 μm ROI were quantified as present or absent adjacent to 2Phatal exposed cells and control nuclei within the same image before, 1 day and 2 days after 2Phatal induction. For quantification of calcium dynamics, time-lapse images were acquired in awake head-fixed mice for three trials (Fig. 5) or six trials (Fig. 7) at 1 Hz for 120 s each trial and at each time point relative to 2Phatal photobleaching (before, 2 h and 6 h for calcium changes during single-cell death and 1 day after on neighbouring cells for spontaneous activity after PV + interneuron ablation). Changes in GCaMP6 fluorescence intensity indicated as $\Delta F/F$ where $F$ is baseline fluorescence intensity within a 10 × 10 μm ROI centred on the cell. Spike events were quantified as fluorescence intensity changes greater than 0.5 change in fluorescence over the average fluorescence for the trial (Figs 5e,f

and 7d,e). For quantification of changes in mitochondria during apoptosis, time-lapse fluorescence images were aligned and thresholded. Line profiles were measured along single dendrites of 2Phatal and control cells as demonstrated (Supplementary Fig. 5). Percent coverage was determined based on the proportion of 0 to non-0 values at any given point along the dendrite (Supplementary Fig. 5d).

Statistical analyses were performed using two-tailed unpaired or unpaired Student's $t$-tests with Holm–Sidak correction where appropriate using Graphpad Prism as indicated in the text and figure legends with significance designated with $P$-values $<0.05$ or by using a 99% confidence interval test (Fig. 2c), or a two-way analysis of variance with Bonferroni $post\ hoc$ analysis (Fig. 8g). For cell survival analysis Log-rank (Mantel–Cox) tests were performed on Graphpad Prism as indicated in the text. All data were assumed to have a normal distribution and equal variance for each statistical test and plotted as the mean ± s.e.m. as indicated in each figure legend. No data were excluded from analysis, no randomization was used to assign experimental subjects and experimenter blinding was not necessary. No statistical methods were used for predetermined sample size determination. For each experiment, at least three animals were used with animal and cell numbers indicated in the text. Each representative image was successfully repeated in excess of at least three image locations for each animal with sample sizes ($n$) designates as single cells followed over multiple days or single animals imaged as indicated in the text and figure legends.

**Data availability.** The data that support the findings of this study are available from the corresponding author on reasonable request.

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

## Acknowledgements

We thank Suk-Won Jin and Jun-Dae Kim (Yale University) for sharing transgenic zebrafish, Akiko Nishiyama (University of Connecticut) for sharing *NG2*cre mice and Flora Vaccarino (Yale University) for sharing Ald1h1L1bacTrap mice. Shannon Leslie (Yale University) helped with naming the technique. This work was supported by the following grants from the National Institutes of Health: R21NS087511, R21NS088411, R01NS0889734 to J.G., and F32NS090820 to R.A.H.

## Author contributions

R.A.H. and J.G. conceived of and designed all experiments. R.A.H. performed all experiments. E.C.D. contributed to the technique characterization. F.C. performed in utero electroporation. A.C.K. contributed transgenic mice and guidance on experimentation. R.A.H. and J.G. wrote the manuscript.

## Additional information

**Competing interests:** The authors declare no competing financial interests.

