## [Peer Review File · Nature Communications]

Reviewers' comments:

Reviewer #1 (Remarks to the Author):

Study of apoptosis process at the single cell level will greatly advance our understanding of the molecular mechanisms of cell death. Especially if the technique can be applied to living animals to study the physiological and behavioral consequences of manipulation of cell death with high spatial and temporal precision, it may be able to replace or complement conventional genetic mice models, which are laborious, expensive and time-consuming. This manuscript reports the novel technique called "2Phatal". This is based on the nuclear apoptosis induced by nucleic acid dye photobleaching. The authors demonstrated the induction of apoptosis at various cell types with highly reproducible manner at the single cell level. Furthermore, they analyzed other cellular processes associated with apoptosis, such as calcium loading, mitochondrial morphology and phagocytosis. The results are clear, the experiments are well designed and the data are promising. However, the reviewer also found the serious limitations of their findings, as described below and those aspects should be considered.

1. My main concern is that DNA-interchelating nuclear dye and its photobleaching is the only cell death inducer used in this study. This can be a good platform to manipulate cell death at the single cell level in a time and region-specific manner at the tissue and animal level. However, its application to the mechanistic understanding of other instances of apoptotic processes may be limited and the authors' claim of allowing mechanistic dissection of general apoptosis process (for example, line 196-198 or line 267-268) better be toned down. The etiology and dynamics of cell death triggered by other cell death inducers can be different. Therefore, 2Phatal seems to have limited implication for the understanding of the biochemical mechanisms of apoptosis *in vivo*.

2. Although 2Phatal may have the potential of being a good platform to study the physiological outcomes of cell type-specific apoptosis at the animal level, it is not demonstrated in this study. It would be necessary to show that 2Phatal can be used to induce physiological outcomes at the organ or animal level by ablating good size of specific population of neurons or other types of cells. If not, 2Phatal is simply a way to induce cell death *in vivo*, one at a single time. How many cells are needed to be ablated to see any consequences at zebra fish? Those data will make 2Phatal more interesting and attractive tool for the study of cell death *in vivo*.

3. Difference in patterns of apoptosis-associated features are truly due to differences in cell types but not due to other factors? Delayed apoptosis in astrocytes needs to be taken with precaution. Astrocytes were co-stained with both SR101 and Hoechst33342, whereas neurons and other cell types were stained with only Hoechst. SR101 may possibly compete with Hoechst for interaction with nuclear DNA and nuclear DNA in astrocytes were not as potently integrated with Hoechst as in neurons. Delayed apoptosis in astrocytes better be confirmed using different astrocyte label method.

Minor comments

1. Figure 5d, there is no signal in 153 s image of normal cells.
2. Fig. 5e was not mentioned.
3. Fig. 6, the arrow was not explained. There are other figures showing arrows without explanation.
4. Line 80. Supplementary Fig. 2, not 1.
5. Line 184 Fig. 5f should be inserted.
6. Line 239-242. Fig. 8 g,i,j were mislabeled. They must be Fig. 8e, f, g.

Reviewer #2 (Remarks to the Author):

This paper reports a single cell ablation approach, which the authors call two-photon chemical apoptotic targeted ablation (2Phatal), in combination with Hoechst 33342 (H33342) and regional scanning of regions of interest by two-photon laser (775 nm). The ablation appears to occur very rapidly in the order of seconds while previous methods were in the orders of minutes and hours. The authors successfully ablated a single cell without affecting neighboring cells in a rapid time course (1-2 days for neuronal death and 4-5 days for the death of astrocytes). With a great deal of data including movies, the authors demonstrate a variety of phenotypes of apoptosis by 2Phatal that include ribosomal disassembly, calcium overloaded, mitochondrial fission in mice and neuromast hair cells in zebrafish. I have little doubt about the technical innovation of 2Phatal. However, the biological evidence for the mechanisms of apoptosis by 2Phatal is still fragmental. For example, they show a new finding which is not previously recognized, ribosomal disassembly at very early stages of apoptosis. It would be desirable for them to pursue the mechanisms for this observation a step further. At the present form of paper each piece of observations that they found by 2Phatal is scattered and there is no integrated story among the pieces. I think that it is important for them to prove that 2Phatal technique can contribute to understanding new mechanisms which has not been probed by other methods, otherwise it is difficult to truly evaluate the value of this technique even though all the data shown are beautiful and impressive.

Major comments

- 1) In Fig 3a, Thy1—GFP positive dendrites, the apparent adjacent axonal boutons of ablated cells show no evidence of alteration. However, there is no direct evidence that the adjacent neuron has direct connections with the ablated cells. It would be better to show that these two cells have been directly connected before 2Phatal by showing physiological experiments and neurochemical methods.
- 2) The picture of Fig 7d suggests that ribosomes were rapidly disassembled 1 day after 2Phatal induction as the authors claim, but this needs to be confirmed by other methods other than just showing the imaging of the Aldh1L1-EGFP bacTRAP transgenic mice after 2Phatal.

Minor comments

There are many descriptions that we can intuitively speculate what authors want to say, but they are not always explicitly clear. These cause ambiguity. I show several such examples.

- 1) Aldh1L1 bacTRAP transgenic mice: it is not clear how this line is constructed and they obtained it. They only cite Aldh1L1 bacTRAP with reference 30 (TRAP) and Aldh1L1cre lines with reference 47. But apparently this is not Aldh1L1 bac TRAP line itself, isn't it.
- 2) Fig 4d: There is no description on what top and bottom panels mean.
- 3) What are the transgenic zebrafish lines of KDR-GFP and Rpox1-RFP made. There is no citation and any description on these lines.
- 4) <df> is probably <degree of freedom>, but no specification for this abbreviation.
- 5) Supplementary Video 6 on line 196 (in PDF file) is not found. What I find is astrocyte apoptotic ribosome dynamics as Supplementary Video 6 in the files attached but not that of mitochondrial dynamics.

Reviewer comments:

Reviewer #1 (Remarks to the Author):

Study of apoptosis process at the single cell level will greatly advance our understanding of the molecular mechanisms of cell death. Especially if the technique can be applied to living animals to study the physiological and behavioral consequences of manipulation of cell death with high spatial and temporal precision, it may be able to replace or complement conventional genetic mice models, which are laborious, expensive and time-consuming. This manuscript reports the novel technique called "2Phatal". This is based on the nuclear apoptosis induced by nucleic acid dye photobleaching. The authors demonstrated the induction of apoptosis at various cell types with highly reproducible manner at the single cell level. Furthermore, they analyzed other cellular processes associated with apoptosis, such as calcium loading, mitochondrial morphology and phagocytosis. The results are clear, the experiments are well designed and the data are promising. However, the reviewer also found the serious limitations of their findings, as described below and those aspects should be considered.

1. My main concern is that DNA-interchelating nuclear dye and its photobleaching is the only cell death inducer used in this study. This can be a good platform to manipulate cell death at the single cell level in a time and region-specific manner at the tissue and animal level. However, its application to the mechanistic understanding of other instances of apoptotic processes may be limited and the authors' claim of allowing mechanistic dissection of general apoptosis process (for example, line 196-198 or line 267-268) better be toned down. The etiology and dynamics of cell death triggered by other cell death inducers can be different. Therefore, 2Phatal seems to have limited implication for the understanding of the biochemical mechanisms of apoptosis in vivo.

We agree with the reviewer and appreciate this concern. We have modified the text to clarify that the strength of our technique is DNA damage-induced cell death at the single cell level in a time and region-specific manner. While our technique is very useful for studying mechanisms of apoptosis *in vivo*, we agree that findings using this technique need to be interpreted cautiously when generalizing them outside of DNA damage to other diverse causes and mechanisms of apoptosis.

2. Although 2Phatal may have the potential of being a good platform to study the physiological outcomes of cell type-specific apoptosis at the animal level, it is not demonstrated in this study. It would be necessary to show that 2Phatal can be used to induce physiological outcomes at the organ or animal level by ablating good size of specific population of neurons or other types of cells. It not, 2Phatal is simply a way to induce cell death in vivo, one at a single time. How many cells are needed to be ablated to see any consequences at zebra fish? Those data will make 2Phatal more interesting and attractive tool for the study of cell death in vivo.

The ability to dissect this process at the single cell level is the distinct advantage of 2Phatal. While ablation of a large number of cells can be useful for dissection of physiological outcomes at the animal level, 2Phatal provides the first method to induce spatiotemporal targeted single cell apoptosis in live animals. Previously it was not possible to induce apoptosis of a single cell and visualize the cell death process followed by the functional consequences to the surrounding microenvironment. Several other methods for large scale cell ablation already exist (diphtheria toxin, caspase 3 or herpes virus) and while it would be easy and feasible to ablate numerous cells using 2Phatal, we feel that experiments such as

functional characterization of circuits after large scale ablation are entirely beyond the scope of this study. Those studies are not essential at all for demonstrating the robustness of 2Phatal as the first method for spatiotemporally targeted single cell ablation in vivo.

3. Difference in patterns of apoptosis-associated features are truly due to differences in cell types but not due to other factors? Delayed apoptosis in astrocytes needs to be taken with precaution. Astrocytes were co-stained with both SR101 and Hoechst33342, whereas neurons and other cell types were stained with only Hoechst. SR101 may possibly compete with Hoechst for interaction with nuclear DNA and nuclear DNA in astrocytes were not as potently integrated with Hoechst as in neurons. Delayed apoptosis in astrocytes better be confirmed using different astrocyte label method.

We appreciate this concern and have performed new experiments to specifically address this question. We have added new data and quantification using GFP-labeled instead of SR101 labeled astrocytes (See Supplementary Figure 6d and text). These new data are consistent with our findings with SR101 dye labeling and provide strong evidence that the difference in apoptotic induction between astrocytes and neurons is not due to an astrocyte SR101 dye labeling artifact.

Minor comments

1. Figure 5d, there is no signal in 153 s image of normal cells.

This image indicates that the GCaMP6s fluorescence signal is very low at baseline. For control cells that are not spiking there is almost no GCaMP6s fluorescence (see all adjacent cells in Figure 5a-b that were not targeted for 2Phatal). In contrast, cells that have been targeted for 2Phatal demonstrate sustained GCaMP6s signals starting approximately 2 hours after the initial photo bleaching. The data can be visualized in Supplementary Video 5 as this is the same control cell shown in the Figure 5d.

2. Fig. 5e was not mentioned.

We have now referenced this panel in the text.

3. Fig. 6, the arrow was not explained. There are other figures showing arrows without explanation.

This has been added to the figure legend and other arrows without explanation have now been referenced in the figure legends.

4. Line 80. Supplementary Fig. 2, not 1.

This has been fixed in the text.

5. Line 184 Fig. 5f should be inserted.

This has now been inserted in the text.

6. Line 239-242. Fig. 8 g,i,j were mislabeled. They must be Fig. 8e, f, g.

This has been fixed in the text.

Reviewer #2 (Remarks to the Author):

This paper reports a single cell ablation approach, which the authors call two-photon chemical apoptotic targeted ablation (2Phatal), in combination with Hoechst 33342(H33342) and regional scanning of regions of interest by two-photon laser (775 nm). The ablation appears to occur very rapidly in the order of seconds while previous methods were in the orders of minutes and hours. The authors successfully ablated a single cell without affecting neighboring cells in a rapid time course (1-2 days for neuronal death and 4-5 days for the death of astrocytes). With a great deal of data including movies, the authors demonstrate a variety of phenotypes of apoptosis by 2Phatal that include ribosomal disassembly, calcium overloaded, mitochondrial fission in mice and neuromast hair cells in zebrafish. I have little doubt about the technical innovation of 2Phatal. However, the biological evidence for the mechanisms of apoptosis by 2Phatal is still fragmental. For example, they show a new finding which is not previously recognized, ribosomal disassembly at very early stages of apoptosis. It would be desirable for them to pursue the mechanisms for this observation a step further. At the present form of paper each piece of observations that they found by 2Phatal is scattered and there is no integrated story among the pieces. I think that it is important for them to prove that 2Phatal technique can contribute to understanding new mechanisms which has no been probed by other methods, otherwise it is difficult to truly evaluate the value of this technique even though all the data shown are beautiful and impressive.

Major comments

1) In Fig 3a, Thy1—GFP positive dendrites, the apparent adjacent axonal boutons of ablated cells show no evidence of alteration. However, there is no direct evidence that the adjacent neuron has direct connections with the ablated cells. It would be better to show that these two cells have been directly connected before 2Phatal by showing physiological experiments and neurochemical methods.

We agree that axonal bouton proximity does not indicate synaptic connectivity. In the initial submission we were careful not to state that these two cells were synaptically connected and were instead using this data to demonstrate that adjacent neural structures were not acutely damaged due to 2Phatal photo bleaching. These data provide strong evidence that this technique could be used in future studies for the exact experiment that the reviewer is requesting but is beyond the scope of our paper.

2) The picture of Fig 7d suggests that ribosomes were rapidly disassembled 1 day after 2Phatal induction as the authors claim, but this needs to be confirmed by other methods other than just showing the imaging of the Aldh1L1-EGFP bacTRAP transgenic mice after 2Phatal.

We appreciate this concern however the bacTRAP mice are well characterized and widely used as a ribosomal specific probe. We are not aware of any other methods that can determine ribosomal disassembly in vivo. Additionally, given that this is a minor claim in our study, further experiments are beyond the scope of this manuscript and reveal the potential of using 2Phatal for numerous studies in the future. We have toned down the statements concerning early astrocytic ribosomal disassembly in our manuscript.

Minor comments

There are many descriptions that we can intuitively speculate what authors want to say, but they are not always explicitly clear. These cause ambiguity. I show several such examples.

1) Aldh1L1 bacTRAP transgenic mice: it is not clear how this line is constructed and they obtained it. They only cite Aldh1L1 bacTRAP with reference 30 (TRAP) and Aldh1L1cre lines with reference 47. But apparently this is not Aldh1L1 bac TRAP line itself, isn't it.

This has been clarified in the methods section under Animals. We have two different Aldh1L1 mouse lines. The Aldh1L1-bacTRAP-EGFP line with ribosomal labeling in astrocytes and the Aldh1L1cre line which provides cytoplasmic GFP labeling of astrocytes when crossed with a GFP reporter (Z/EG in this case, Supplementary Figure 6).

2) *Fig 4d: There is no description on what top and bottom panels mean.*

This has been added to the figure legend.

3) *What are the transgenic zebrafish lines of KDR-GFP and Rpox1-RFP made. There is no citation and any description on these lines.*

Citations and sources have been added to the text.

4) *<df> is probably <degree of freedom>, but no specification for this abbreviation.*

This has been added to the figure legends.

5) *Supplementary Video 6 on line 196 (in PDF file) is not found. What I find is astrocyte apoptotic ribosome dynamics as Supplementary Video 6 in the files attached but not that of mitochondrial dynamics.*

This has been corrected in the text.

Reviewers' comments:

Reviewer #1 (Remarks to the Author):

The authors addressed well most of the points raised in the first round of review. However, I think that it is important for them to demonstrate in vivo functional outcome by inducing cell death using 2Phtal technique. Otherwise, it is simply a technique to cause single cell death at the tissue with very limited implications. In the Abstract, they claimed that "a major bottleneck limiting understanding of mechanisms and consequences of cell death in complex organism is the inability to induce and visualize this process with spatial and temporal precision in living animals." I agree that this 2Phtal technique is really a good method to induce and visualize cell death process in living animals by inducing cell death at single cell level with high precision. However, as mentioned in the first round of review, this is limited to Hoechst dye-intercalating DNA damage-induced apoptosis. Therefore, contribution to understanding of cell death mechanisms is still restricted to related types of apoptosis. Detailed mechanisms of cell death can be studied in vitro and there are many other options, but what makes 2Phtal method more valuable will be application to in vivo functional study of cell death by ablating targeted groups of cells without widespread, non-specific cell death. I strongly think that lack of tools to induce controlled cell death with high precision to study physiological consequences in vivo is more serious bottleneck. Again, quoting the authors' claims in the Abstract, they acclaimed that "2Phtal provides a powerful and rapidly adoptable platform to investigate in vivo functional consequences and neural plasticity following cell death as well as apoptosis, cell clearance and tissue remodeling in diverse organs and species." However, they did not provide the data showing that 2Phtal can be useful for those applications of "in vivo functional consequences" or "tissue remodeling". I am impressed with their hard work and beautiful data, but am sorry to hear that scaling up of cell death using 2Phtal to see functional outcomes at the organ or animal level is not included in their plan. I strongly feel those data are necessary for final acceptance.

Reviewer #2 (Remarks to the Author):

The revised manuscript is well improved. If this is to be accepted, the following points should be corrected.

1. Line 83: (Fig. 1 a) should be (Fig. 1b)
2. Line 135: Because the authors do not show the two cells are connected and Figure 3 pictures were only taken from a limited angle, it is difficult to judge how close cell boutons of green color and a red cell are. Therefore, "immediately" should be deleted.
3. Line 236: (Fig. 9a) should be (Fig. 8a)

Reviewer comments:

Reviewer #1 (Remarks to the Author):

The authors addressed well most of the points raised in the first round of review. However, I think that it is important for them to demonstrate in vivo functional outcome by inducing cell death using 2Phtal technique. Otherwise, it is simply a technique to cause single cell death at the tissue with very limited implications. In the Abstract, they claimed that “a major bottleneck limiting understanding of mechanisms and consequences of cell death in complex organism is the inability to induce and visualize this process with spatial and temporal precision in living animals.” I agree that this 2Phtal technique is really a good method to induce and visualize cell death process in living animals by inducing cell death at single cell level with high precision. However, as mentioned in the first round of review, this is limited to Hoechst dye-intercalating DNA damage-induced apoptosis. Therefore, contribution to understanding of cell death mechanisms is still restricted to related types of apoptosis. Detailed mechanisms of cell death can be studied in vitro and there are many other options, but what makes 2Phtal method more valuable will be application to in vivo functional study of cell death by ablating targeted groups of cells without widespread, non-specific cell death. I strongly think that lack of tools to induce controlled cell death with high precision to study physiological consequences in vivo is more serious bottleneck. Again, quoting the authors’ claims in the Abstract, they acclaimed that “2Phtal provides a powerful and rapidly adoptable platform to investigate in vivo functional consequences and neural plasticity following cell death as well as apoptosis, cell clearance and tissue remodeling in diverse organs and species.” However, they did not provide the data showing that 2Phtal can be useful for those applications of "in vivo functional consequences" or "tissue remodeling". I am impressed with their hard work and beautiful data, but am sorry to hear that scaling up of cell death using 2Phtal to see functional outcomes at the organ or animal level is not included in their plan. I strongly feel those data are necessary for final acceptance.

We appreciate this concern and while this technique is not designed for experiments at the organ or whole animal level we understand the need for functional readouts to demonstrate the utility of 2Phatal for future studies. To address this, we have now included a new proof of principle experiment to test the functional consequence of 2Phatal ablation of a targeted population of neurons. This experiment improves the paper and nicely complements the characterization of this technique and we thank the reviewer for the suggestion.

In this new experiment we demonstrate that 2Phatal ablation of 8-11 fast spiking interneurons resulted in increased spontaneous activity of the remaining neighboring neurons (Figure 7 and described on Page 4 of the revised manuscript). To our knowledge this is the first demonstration of changes in local activity after targeted neuron ablation in vivo, an experiment which is now feasible using 2Phatal. These data indeed show a functional consequence and demonstrate that future studies can now dissect the precise cellular and molecular mechanisms involved, including plasticity in the microcircuit over time and begin to ask what happens to these circuits after 2Phatal of numerous other cell types.

Importantly, these new data establish the potential impact and utility of this technique and demonstrate the immediate questions that can be asked for broad exploration of the functional consequences and tissue remodeling after targeted cell ablation at the microcircuit level for the first time in vivo.

Reviewer #2 (Remarks to the Author):

The revised manuscript is well improved. If this is to be accepted, the following points should be corrected.

1. Line 83: (Fig. 1 a) should be (Fig. 1b)

We have corrected this in the revised text.

2. Line 135: Because the authors do not show the two cells are connected and Figure 3 pictures were only taken from a limited angle, it is difficult to judge how close cell boutons of green color and a red cell are. Therefore, “immediately “ should be deleted.

We have changed this in the revised text.

3. Line 236: (Fig. 9a) should be (Fig. 8a)

We have changed this in the revised text.

REVIEWERS' COMMENTS:

Reviewer #1 (Remarks to the Author):

The authors addressed the critiques well and there is no further concern.

REVIEWERS' COMMENTS:

Reviewer #1 (Remarks to the Author):

The authors addressed the critiques well and there is no further concern.

We thank the reviewer for their previous suggestions and comments.